



# Secondary PM decreases significantly less than NO₂ emission reductions during COVID lockdown in Germany

Vigneshkumar Balamurugan[1], Jia Chen[1], Zhen Qu[2], Xiao Bi[1], and Frank N. Keutsch[2,3]

[1]Environmental Sensing and Modeling, Technical University of Munich (TUM), Munich, Germany
[2]School of Engineering and Applied Science, Harvard University, Cambridge, MA, USA
[3]Department of Chemistry and Chemical Biology, Harvard University, Cambridge, MA, USA

**Correspondence:** Vigneskumar Balamurugan (vigneshkumar.balamurugan@tum.de), Jia Chen (jia.chen@tum.de)

**Abstract.** This study estimates the influence of anthropogenic emission reductions on the concentration of particulate matter with a diameter smaller than 2.5 $\mu$m (PM$_{2.5}$) during the 2020 lockdown period in German metropolitan areas. After accounting for meteorological effects, PM$_{2.5}$ concentrations during the spring 2020 lockdown period were 5 % lower compared to the same time period in 2019. However, during the 2020 pre-lockdown period (winter), meteorology accounted for PM$_{2.5}$

concentrations were 19 % lower than in 2019. Meanwhile, meteorology accounted for NO$_2$ concentrations dropped by 23 % during the 2020 lockdown period compared to an only 9 % drop for the 2020 pre-lockdown period, both compared to 2019. Meteorology accounted for SO$_2$ and CO concentrations show no significant changes during the 2020 lockdown period compared to 2019. GEOS-Chem (GC) simulation with a COVID-19 emission reduction scenario based on the observations (23 % reduction in NO$_X$ emission with unchanged VOC and SO$_2$) are consistent with the small reductions of PM$_{2.5}$ during the

lockdown and are used to identify the underlying drivers for this. Due to being in a NO$_X$ saturated ozone production regime, GC OH radical and O$_3$ concentrations increased (15 and 9 %, respectively) during the lockdown compared to a Business As Usual (no lockdown) scenario. The increased O$_3$ results in increased NO$_3$ radical concentrations, primarily during the night, despite the large reductions in NO$_2$. Thus, the oxidative capacity of the atmosphere is increased in all three important oxidants, OH, O$_3$, and NO$_3$. PM nitrate formation from gas-phase nitric acid (HNO$_3$) is decreased during the lockdown as the increased

OH concentration cannot compensate for the strong reductions in NO$_2$ resulting in decreased day-time HNO$_3$ formation from the OH + NO$_2$ reaction. However, night-time formation of PM nitrate from N$_2$O$_5$ hydrolysis is relatively unchanged. This results from the fact that increased night-time O$_3$ results in significantly increased NO$_3$ which roughly balances the effect of the strong NO$_2$ reductions on N$_2$O$_5$ formation. Ultimately, the only small observed decrease in lockdown PM$_{2.5}$ concentrations can be explained by the large contribution of night-time PM nitrate formation, generally enhanced sulfate formation

and slightly decreased ammonium. This study also suggests that high PM$_{2.5}$ episodes in early spring are linked to high atmospheric ammonia concentrations combined with favorable meteorological conditions of low temperature and low boundary layer height. North-West Germany is a hot-spot of NH$_3$ emissions, primarily emitted from livestock farming and intensive agricultural activities (fertilizer application), with high NH$_3$ concentrations in the early spring and summer months. Based on our findings, we suggest that appropriate NO$_X$ and VOC emission controls are required to limit ozone, and that should also





help reduce PM$_{2.5}$. Regulation of NH$_3$ emissions, primarily from agricultural sectors, could result in significant reductions in PM$_{2.5}$ pollution.

# 1 Introduction

To halt the spread of the COVID-19 virus, various strict measures such as social isolation, curfews, and travel restrictions were implemented around the world in early 2020 (Steinmetz et al., 2020). As a result of these restrictions, anthropogenic emissions
decreased significantly (Schumann et al., 2021; Le Quéré et al., 2020; Turner et al., 2020). Reduced primary emission activities from road transportation and industrial activities were expected to improve air quality. Numerous studies using satellite and in-situ measurements have reported significant reductions in primary air pollutant concentrations during the COVID-19 lockdown period compared to pre-lockdown period in various parts of the world (Bauwens et al., 2020; Biswal et al., 2020; Collivignarelli et al., 2020; Dietrich et al., 2021; Field et al., 2020; He et al., 2021; Pathakoti et al., 2020; Mendez-Espinosa et al., 2020), but
also emphasize the importance of accounting for the effects of different meteorological conditions between the study period and the reference period (Barré et al., 2020; Grange et al., 2020; Kroll et al., 2020; Koukouli et al., 2021; Ordóñez et al., 2020; Solberg et al., 2021). Anomalies in air pollutant concentrations caused by changes in meteorological conditions were also separated from observed changes using modeling work to estimate the actual influence of COVID-19 lockdown restrictions on air pollutant concentration changes (Balamurugan et al., 2021; Goldberg et al., 2020; Kang et al., 2020; Petetin et al., 2020; Qu
et al., 2021; Yin et al., 2021). Secondary pollutant concentrations (O$_3$ and PM$_{2.5}$), which are primarily produced by precursor gases through complex atmospheric chemical reactions, remarkably increased or did not reduce commensurate to precursor emission reductions seen in some parts of the world during the COVID-19 lockdown period (Campbell et al., 2021; Deroubaix et al., 2021; He et al., 2021; Huang et al., 2021; Keller et al., 2021; Lee et al., 2020; Putaud et al., 2021; Souri et al., 2021; Wang et al., 2020, 2021).

Particulate Matter (PM) is the sum of all particles (solid and liquid) suspended in air, and can be classified based on aerodynamic behavior, i.e., aerodynamic diameter (AD). Particles with an AD smaller than 10 $\mu$m are referred to as PM$_{10}$, while particles smaller than 2.5 $\mu$m AD are referred to as PM$_{2.5}$. Understanding of seasonal and inter-annual variability of PM, particularly over urban areas, remains a challenge (Fuzzi et al., 2015). This is mainly due to a lack of understanding in the attribution of PM sources. PM sources include both direct/primary sources (vehicle and industrial emissions, wind-blown dust,
pollen, wildfires, etc.) as well as secondary formation (gas-to-particle conversion process) via atmospheric chemical reaction of precursor compounds such as NO$_X$, SO$_2$, NH$_3$, VOCs and other organic compounds, including compounds that have partitioned from primary aerosol back to the gas-phase, followed by partitioning to the condensed phase (Allen et al., 2015; Ayres et al., 2015; Fisher et al., 2016; Hallquist et al., 2009; Jacob, 1999; Jacobson, 1999; Marais et al., 2016; Seinfeld and Pankow, 2003; Steinfeld, 1998; Zhang et al., 2015). The composition of PM thus varies greatly depending on time and location; for
example, in urban areas nitrate and organic aerosol often dominate in winter time (Cesari et al., 2018; Zhai et al., 2021).

In this study, we mainly focus on the response of urban surface PM$_{2.5}$ to COVID-19 lockdown restrictions in Germany. Because major anthropogenic emissions are reduced, this unplanned intervention can test the understanding of the contribu-





tion of secondary $PM_{2.5}$ sources, as well as the processes important in secondary $PM_{2.5}$ formation. Despite of significant reductions in some anthropogenic activities, natural and agricultural air pollutant sources were not affected by the COVID-19

lockdown measures. Ammonia ($NH_3$) emissions (agricultural sources) are a significant source of $PM_{2.5}$ in Germany in the spring (Fortems-Cheiney et al., 2016), when lockdown restrictions are implemented. Secondary inorganic aerosols such as ammonium sulfate and ammonium nitrate are the largest contributors to $PM_{2.5}$ in Europe (Pay et al., 2012; Petetin et al., 2016). In comparison to sulfate formation, nitrate formation is more dependent on $NH_3$ concentration (Erisman and Schaap, 2004; Sharma et al., 2007; Wu et al., 2008). In the winter and spring (low temperature and high relative humidity), the role of $NH_3$ in

$PM_{2.5}$ formation is greater than in the summer (high temperature and low relative humidity) (Schiferl et al., 2016; Squizzato et al., 2013; Viatte et al., 2020). Primary components of $PM_{2.5}$ are directly proportional to primary emission but secondary components of $PM_{2.5}$ are not directly proportional to secondary precursor emissions or concentrations as they are produced by non-linear complex atmospheric chemical reactions (Shah et al., 2018). Observational and modeling evidence is required to estimate the influence of change in precursor emissions on $PM_{2.5}$ concentrations. To this end, we used ground and space-based

measurements of $PM_{2.5}$, $NO_2$, $O_3$, $SO_2$, CO and $NH_3$ in conjunction with GEOS-Chem simulations to investigate the influence of lockdown restrictions on $PM_{2.5}$ concentrations.

Modelling studies such as Gaubert et al. (2021); Hammer et al. (2021); Matthias et al. (2021); Menut et al. (2020) have already reported the $PM_{2.5}$ changes across Europe including Germany, during the COVID-19 lockdown period. The activity data (e.g., transportation, industrial activities and energy production) were used in the above mentioned studies to create a COVID-

19 emission reduction scenario (Doumbia et al., 2021; Guevara et al., 2021). However, there are large discrepancies between various activity data sets (Gensheimer et al., 2021), necessitating different approaches to estimating the actual emission reduction caused by the COVID-19 lockdown restrictions. In this study, GEOS-Chem simulations (using identical anthropogenic emission for 2020 and 2019) were used to estimate the meteorology accounted for observed pollutant concentrations changes between 2020 and 2019, which were then used as a proxy for emissions reductions caused by COVID-19 lockdown measures

to create a COVID-19 emission scenario in GEOS-Chem model for simulating the lockdown pollutant concentrations (Fig. 1). In addition to looking at the impact of lockdown restrictions on air pollutant concentrations (Sect. 4.1), we focus on process level analysis of the impact of changes in precursor emissions ($NO_X$) on $PM_{2.5}$ formation (Sect. 4.2), as well as the role of ammonia ($NH_3$) emissions in $PM_{2.5}$ formation (Sect. 4.3).

## 2 Data and Model

Data sets used in this study are summarized in Table 1. We focused on ten metropolitan areas in Germany (Bremen, Cologne, Dresden, Dusseldorf, Frankfurt, Hamburg, Hanover, Leipzig, Munich and Stuttgart) and used surface air pollutant concentration data ($PM_{2.5}$, $NO_2$, $O_3$) for all of these while $SO_2$ data was only available for five of these areas (Bremen, Dresden, Frankfurt, Hamburg and Leipzig) and CO data was limited to six metropolitan areas (Bremen, Frankfurt, Hamburg, Hanover, Munich and Stuttgart). We use data for 2019 and 2020 in this work (data-obtained from https://discomap.eea.europa.eu/map/fme/

AirQualityExport.htm).





**Table 1.** Data sets used in this study.

| Data source | Data | Temporal resolution | Spatial resolution | Data availability |
|---|---|---|---|---|
| Governmental in-situ measurements | $NO_2$, $O_3$, $PM_{2.5}$ | 1 h | - | Bremen, Cologne, Dresden, Dusseldorf, Frankfurt, Hamburg, Hanover, Leipzig, Munich and Stuttgart metropolitan areas |
| | $SO_2$ | 1 h | - | Bremen, Dresden, Frankfurt, Hamburg and Leipzig metropolitan areas |
| | CO | 1 h | - | Bremen, Frankfurt, Hamburg, Hanover, Munich and Stuttgart metropolitan areas |
| TROPOMI satellite measurements | $SO_2$ | daily | 7*3.5 km (5.5*3.5 km, after August 6, 2019) | All of Germany |
| IASI satellite measurements | $NH_3$ | twice a day | 12 km diameter | All of Germany |
| | | monthly | 1 degree | All of Germany |
| ERA 5 (ECMWF reanalysis) | Temperature, relative humidity, boundary layer height and wind speed | 1 h | 0.25 degree | All of Germany |
| | Precipitation | daily | 1 degree | All of Germany |
| GEOS-Chem (GC) chemical transport model | All species | 1 h | 0.5 * 0.625 degree | All of Germany |

TROPOMI tropospheric $SO_2$ (Theys et al., 2017) column products are also used (offline products-obtained from https://s5phub.copernicus.eu). The TROPOMI $SO_2$ product provides the total $SO_2$ column between the surface and the top of troposphere. The TROPOMI overpass occurs around 13.30 local time. At the start of the mission, the TROPOMI product provided data at a resolution of 7*3.5 km, while after August 6, 2019 the resolution improved to 5.5*3.5 km. Stricter quality filtering

criteria (quality assurance value (qa) >= 0.5) was applied to the dataset. A daily mean of $SO_2$ is calculated by averaging these values within 0.5-degree radius of the urban center.

The daily atmospheric $NH_3$ variability in Germany was studied using the "near-real time daily IASI/Metop-B ammonia (NH3) total column (ANNI-NH3-v3)" dataset (products-obtained from https://iasi.aeris-data.fr/catalog/). The data used are from the IASI instrument aboard the Metop-B satellite, which has a local solar overpass time of 9:30 a.m and 9:30 p.m

(Clerbaux et al., 2009). We only used day-time (9.30 am) measurements in this study. Night-time measurements (9.30 pm) were excluded due to their large relative errors. A daily mean is calculated by averaging the values within 0.5-degree radius of the urban center. The monthly atmospheric $NH_3$ variability in Germany was studied using the "standard monthly IASI/Metop-B ULB-LATMOS ammonia (NH3) L3 product (total column)" dataset. This product contains a monthly averaged $NH_3$ total column with a spatial resolution of 1*1 degree (products-obtained from https://iasi.aeris-data.fr/catalog/).

Temperature, relative humidity, boundary layer height and wind information are obtained from the ERA 5 product (Hersbach et al., 2020). This product's native spatial and temporal resolutions are 0.25 degree and 1 hour, respectively. For precipitation information, the GPCP daily gridded product from ERA 5 is used, which provides global gridded data at 1-degree resolution (products-obtained from https://cds.climate.copernicus.eu/).

We used the GEOS-Chem (GC) chemical transport model (http://doi.org/10.5281/zenodo.3959279) to simulate the pollutant

concentration for 2020 and 2019. The GC simulation conducted over Germany (4-17°E, 45-57°N) had a horizontal resolution of 0.5°*0.625° with dynamic boundary conditions generated from a global simulation with 4°*5° resolution. We ran the GC simulation for two cases. In the first case, anthropogenic emissions from the 2014 CEDS inventory (Hoesly et al., 2018)





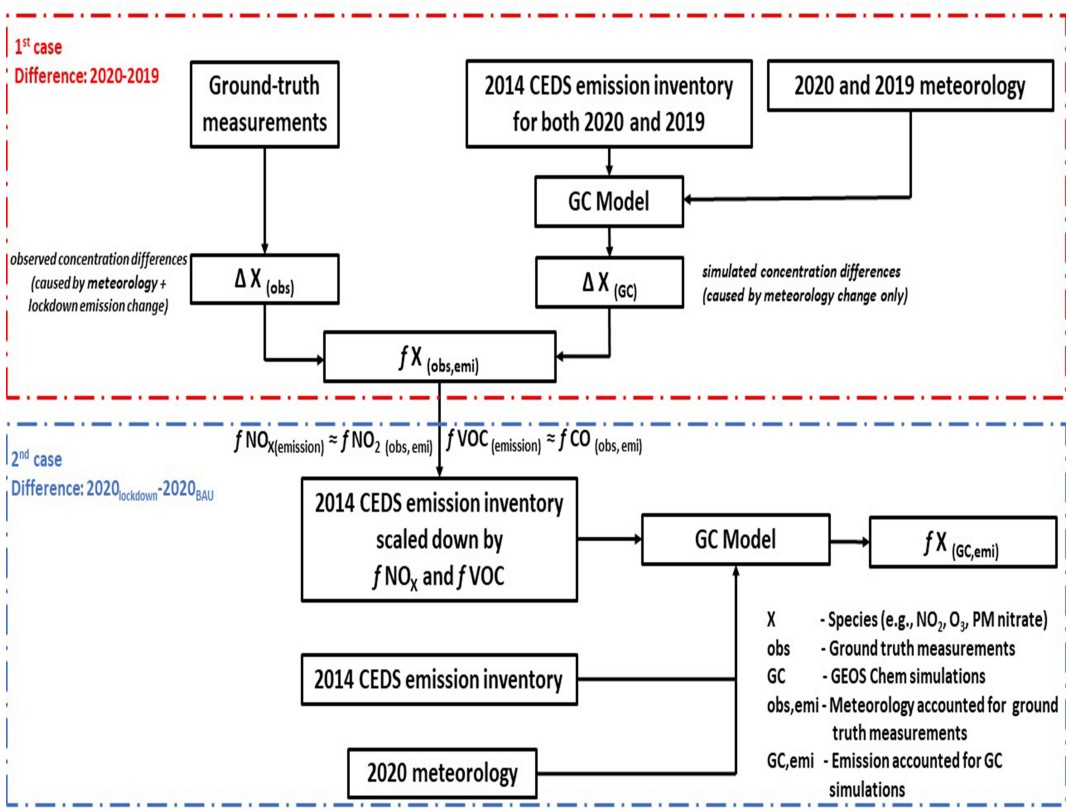

**Figure 1.** Schematic diagram of our methodology for calculating the meteorology accounted for observed pollutant concentrations changes between 2020 and 2019, and emission accounted for GC pollutant concentrations changes between 2020 lockdown and 2020 BAU scenario.

are used in the GC simulations for both 2019 and 2020, but with the corresponding meteorology from MERRA-2 global reanalysis product for 2019 and 2020. Natural emissions from soil and lightning are calculated for the corresponding year using mechanisms described in Hudman et al. (2012) and Murray (2016). The corresponding year's open fire emissions from GFED4 (Werf et al., 2017) are used for 2019 and 2020. In the second case, the anthropogenic emission inventory were scaled down by the estimated emissions reduction caused by the lockdown restrictions for the 2020 lockdown period. The remaining (natural and fire) emissions are calculated in the same way as in the first case.

## 3  Method

The following is our methodology for estimating meteorology accounted for observed pollutant concentration changes between 2020 and 2019, similar to Balamurugan et al. (2021); Qu et al. (2021). We estimate the difference in pollutant concentrations



between 2020 and 2019 caused by changes in meteorology using GC simulated concentrations (first case). Since GC uses identical anthropogenic emission for 2020 and 2019, with the corresponding year meteorology, the difference between 2020 and 2019 GC pollutant (e.g., $PM_{2.5}$) concentrations only results from meteorology changes between 2020 and 2019. We use

$\Delta$ to signify absolute concentration change, and $f$ to signify fractional (percentage) change.

$$\Delta PM_{2.5(GC)} = PM_{2.5(GC,2020)} - PM_{2.5(GC,2019)} \tag{1}$$

The observed (ground-truth measurements) pollutant concentration changes between 2020 and 2019, which includes the effects of lockdown restrictions and meteorology, is:

$$\Delta PM_{2.5(obs)} = PM_{2.5(obs,2020)} - PM_{2.5(obs,2019)} \tag{2}$$

To disentangle the meteorology contribution from the observed pollutant concentration changes, we subtract the GC pollutant concentration changes caused by meteorology from observed pollutant concentration changes between 2020 and 2019.

$$\Delta PM_{2.5(obs,emi)} = \Delta PM_{2.5(obs)} - \Delta PM_{2.5(GC)} \tag{3}$$

The fractional change in meteorology accounted for pollutant concentration between 2020 and 2019, i.e., pollutant concentration changes between 2020 and 2019 due to emission changes only, is calculated as,

$$fPM_{2.5(obs,emi)} = \frac{\Delta PM_{2.5(obs,emi)}}{PM_{2.5(obs,2019)}} \tag{4}$$

where, "obs", "GC" and "obs,emi" refer to ground-truth measurements, GEOS-Chem simulations and meteorology accounted for ground-truth measurements, respectively.

We estimate the meteorology accounted for fractional change in other pollutant concentrations analogously. Our previous study (Balamurugan et al., 2021), using the same methodology, reported the meteorology accounted for $NO_2$ and $O_3$ concen-

tration changes for eight German metropolitan areas. Here, we reproduce the results for $NO_2$ and $O_3$ concentrations, but for ten metropolitan areas. We use $fNO_{2(obs,emi)}$ and $fCO_{(obs,emi)}$ to capture fractional changes in anthropogenic $NO_X$ and VOC emission ($fNO_{X(emission)}$) and $fVOC_{(emission)}$)) due to lock down restrictions, respectively. Because of the scarcity of VOC measurements, CO data was used as a proxy for anthropogenic VOC (Fujita et al., 2003; Jiménez et al., 2005; Stephens et al., 2008; Yarwood et al., 2003) and $NO_2$ was used as proxy for $NO_X$. This assumption is supported by studies such as Baker et al.

(2008); Von Schneidemesser et al. (2010), which show anthropogenic VOC is well correlated with CO, and Blanchard and Tanenbaum (2003), which shows comparable changes in VOC and CO between weekday and weekend. Changes in biogenic VOCs are not directly affected by lockdown measures.

$$fNO_{X(emission)} \approx fNO_{2(obs,emi)} \tag{5}$$

$$fVOC_{(emission)} \approx fCO_{(obs,emi)} \tag{6}$$





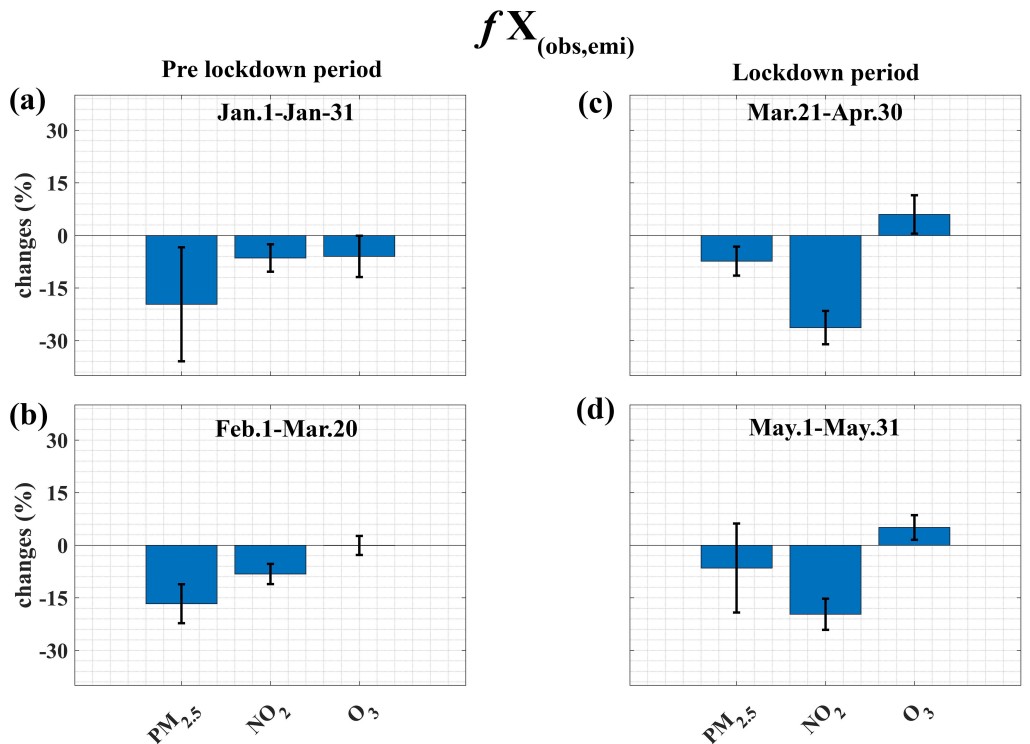

**Figure 2.** Meteorology accounted for mean in-situ $PM_{2.5}$, $NO_2$, and $O_3$ concentration changes between 2020 and 2019. Results of computations according to our first case ($fX_{(obs,emi)}$) in the Sect. 3. Error bars represent the $1\,\sigma$ of mean of ten metropolitan areas.

The base anthropogenic emission inventory were then scaled down by $fNO_{X(emission)}$ and $fVOC_{(emission)}$ for $NO_X$ and VOC emission, respectively, in the GC model for the 2020 lockdown period (second case), which simulates all pollutants concentrations for the lockdown emission scenario. The fractional change in emission accounted for, i.e. using scaled emission inventories, GC pollutants level during the 2020 lockdown period compared to 2020 Business As Usual (BAU), i.e., no

lockdown, level is calculated as,

$$fPM_{2.5(GC,emi)} = \frac{PM_{2.5(GC,2020,lock)} - PM_{2.5(GC,2020)}}{PM_{2.5(GC,2020)}} \tag{7}$$

where, "GC,emi" refers to GC simulations accounting for scaled emission and $PM_{2.5(GC,2020,lock)}$ are the $PM_{2.5}$ concentrations during the lockdown period determined via the 2020 GC simulations with down-scaled emissions. We estimate the emission accounted for concentration changes of other pollutants in the same way. Figure 1 illustrates our methodology for calculat-

ing the meteorology accounted for observed pollutant concentrations changes between 2020 and 2019, as well as emission accounted for GC pollutant concentration changes between 2020 lockdown and 2020 BAU scenario.





## 4  Results and discussion

### 4.1  Influence of lockdown restrictions on the concentrations of air pollutants

To assess the impact of lockdown restrictions on the concentration of air pollutants, we compared the 2020 lockdown period

pollutant concentrations to the same period in 2019. These comparison results, however, need to take the effects of both meteorological and lockdown restrictions into account. As mentioned in Sect. 3, we used GEOS-Chem simulations to disentangle the effects of meteorology on observed pollutant concentration changes between 2020 and 2019. Studies such as Balamurugan et al. (2021) and Tai et al. (2012) have shown that GEOS-Chem can reproduce the temporal variability of observed pollutant concentrations including $PM_{2.5}$, emphasizing that GC can be used for process level analysis of $PM_{2.5}$ variability. We also

compared the 2019 GC and 2019 observed in-situ $PM_{2.5}$ concentrations and found that the GC and observed in-situ $PM_{2.5}$ concentrations were in good agreement (R > 0.5 for all metropolitan areas, except Leipzig which has a R value of 0.39) (e.g., Fig. 6 (c), for Cologne metropolitan area). The GC simulations underestimate the $PM_{2.5}$ when compared to observed in-situ $PM_{2.5}$ concentrations (mean bias (GC - in-situ) ranges from -12.7 % to -37.4 %), except for the Cologne metropolitan area (+ 11.7 %). However, since we use the GC's relative difference between 2020 and 2019, this bias should cancel out.

Figure 2 shows meteorology accounted for mean $PM_{2.5}$, $NO_2$ and $O_3$ concentration changes between 2020 and 2019 for ten German metropolitan areas from January 1 through May 31. Both meteorology accounted and unaccounted for mean $PM_{2.5}$, $NO_2$ and $O_3$ concentration changes between 2020 and 2019 for ten German metropolitan areas are shown in Appendix Fig. A1. The German government imposed COVID-19 lockdown restrictions on March 21, 2020 in Germany. In figures and for specific cases, the pre-lockdown period (January 1 to March 20) is divided into two sections, and the lockdown period

(March 21 to May 31) is also divided into two sections (unless otherwise specified): (a) January 1 to January 31, 2020 - No lockdown restrictions, (b) February 1 to March 20, 2020 - No lockdown restrictions in the event of unusual weather conditions (occurrence of storms), (c) March 21 to April 30, 2020 (spring) - Strict lockdown measures, and (d) May 1 to May 31, 2020 (late spring) - Loose lockdown measures. Germany experienced high wind conditions due to storms in February 2020 (Matthias et al., 2021), which was used to determine the extent of meteorology's role in pollutant concentration changes. Meteorology

unaccounted for mean $NO_2$ and $PM_{2.5}$ concentrations for February 1 to March 20, 2020 period (before the implementation of lockdown) are lower than the corresponding ones in 2019 by 30 % and 42 % ($fNO_{2(obs)}$ and $fPM_{2.5(obs)}$), respectively, due to the dilution/dispersion from the high wind conditions. However, after accounting for meteorology, the difference in mean $NO_2$ and $PM_{2.5}$ concentrations between 2020 and 2019 for the period February 1 to March 20 ($fNO_{2(obs,emi)}$ and $fPM_{2.5(obs,emi)}$) are 8 % and 18 %, respectively. This finding is consistent with meteorology accounted for mean $NO_2$ and $PM_{2.5}$ changes

between 2020 and 2019 for the period January 1 to January 31 (Fig. 2 (a,b)). This highlights the importance of accounting for meteorological impacts.

In the 2020 pre-lockdown period (January 1 to March 20), both meteorology accounted for mean $NO_2$ and $PM_{2.5}$ levels are lower by 9 % and 19 %, respectively, compared to the same period in 2019. During the 2020 lockdown period (March 21 to May 31), mean meteorology accounted for $NO_2$ concentrations dropped significantly (23 %) compared to the same period in 2019,

which is greater than the drop in the 2020 pre-lockdown period compared to 2019 (9 %). Comparatively, mean meteorology





accounted for 2020 lockdown $PM_{2.5}$ concentrations show a smaller reduction (5 %) compared to the same period in 2019, while an important precursor, $NO_2$, decreased by 23 % during the same period. Furthermore, the meteorology accounted for $PM_{2.5}$ reduction during the 2020 lockdown period (5 %) is less than the meteorology accounted for $PM_{2.5}$ reduction observed during the 2020 pre-lockdown period (19 %) compared to the corresponding 2019 periods (Fig. 2). Especially in Munich and

Stuttgart, meteorology accounted for $PM_{2.5}$ concentrations during the 2020 lockdown period are higher than in 2019. The meteorology accounted for mean $O_3$ concentrations in the 2020 lockdown period are increased by 6 % compared to the same period in 2019. The increase in $O_3$ concentration during the 2020 lockdown period is mainly due to being in a $NO_X$ saturated regime (Gaubert et al., 2021), in which reducing $NO_X$ emission results in an increase in $O_3$ concentrations (Sillman, 1999; Sillman et al., 1990).

The effects of lockdown restrictions on $SO_2$ concentrations are insignificant. In comparison to 2019, TROPOMI meteorology accounted for $SO_2$ levels are decreased by 1 % during the 2020 lockdown period compared to 2019 (Fig. A1). For accounting meteorological impacts on TROPOMI satellite column concentrations, GEOS-Chem diagnostics (47 vertical layers) were converted to a column, applying TROPOMI's averaging kernel. Because of the large influence of background concentration on satellite column measurements, we also investigated in-situ $SO_2$ concentrations, but only for five metropolitan areas. Similarly,

we found that the impact of lockdown restrictions on in-situ $SO_2$ concentrations is marginal (Fig. B1). The road transportation sector contributes less than 1 % of total sulfur dioxide emissions, while coal-related fuel burning (industrial and energy production) accounts for nearly 80 % of total sulfur dioxide emissions (SO2, 2021). Because the lockdown restrictions primarily reduced traffic-related emissions, we see far less effects of the lockdown on $SO_2$ concentration (slight increase or no significant decrease in other European metropolitan areas (Collivignarelli et al., 2020; Filonchyk et al., 2021; Higham et al., 2021)). We

found similar effects on in-situ CO concentration changes in six metropolitan areas. The meteorology accounted for mean CO concentrations are lower by 3 % during the 2020 lockdown period compared to 2019 (Fig. B1). Stuttgart meteorology accounted for CO concentrations in 2020 were higher than 2019 at all times. Other metropolitan areas experienced minor reductions (Clark et al., 2021; Hörmann et al., 2021).

### 4.2 Model evidence of changes in air pollutants concentration resulting from lockdown restrictions

As mentioned in Sect. 3, we use the meteorology accounted for $NO_2$ and CO changes to adjust the anthropogenic $NO_X$ and VOC emissions in inventories due to lockdown restriction impacts. GC model simulations are then obtained with this scaled anthropogenic emission scenario (23 % reduction in $NO_X$ emission and unchanged VOC emissions) for the 2020 lockdown period. The $NO_X$ emission reduction is within the range of estimated $NO_X$ emission reductions using activity data for Europe by previous authors (Doumbia et al., 2021; Guevara et al., 2021) (25 % and 33 %, respectively). For those studies there are

large differences in estimated VOC emission changes for Europe; Doumbia et al. (2021) estimated 34 % while Guevara et al. (2021) estimated 8 % reduction in VOC emissions. However, the real-time measurements at a United Kingdom station show no significant changes in many VOC concentrations during the lockdown period (Grange et al., 2020). For the $NO_X$ saturated ozone production regime regime, VOC emission reductions can decrease ozone levels, while $NO_X$ emission reductions increase them. Gaubert et al. (2021) conducted a sensitivity study of modelling work on ozone levels in response to the $NO_X$ or VOC



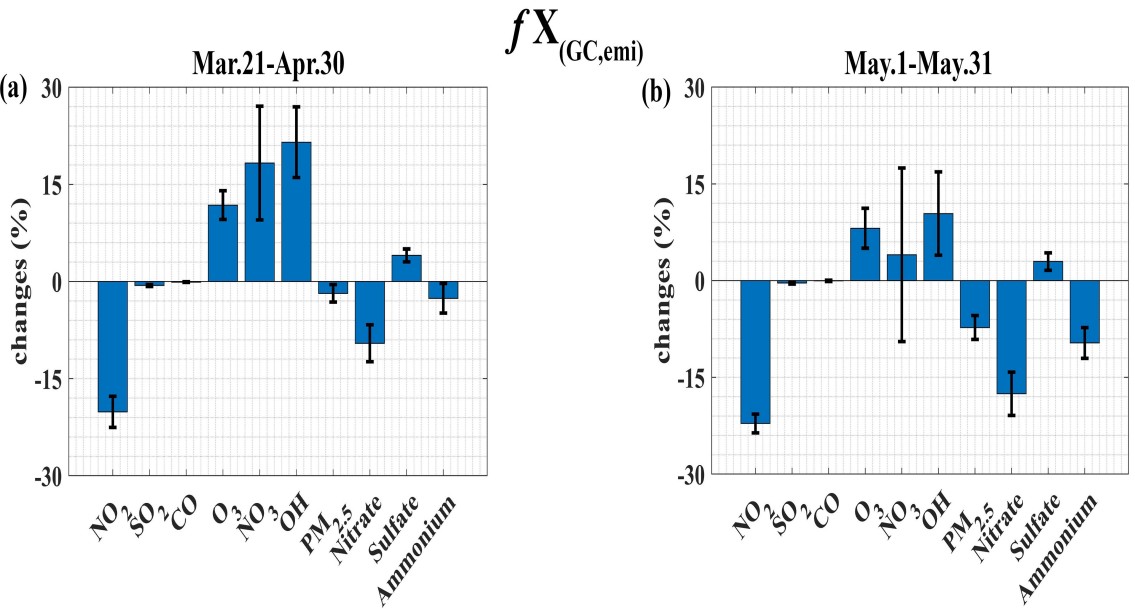

**Figure 3.** The emission accounted for GC $NO_2$, $SO_2$, CO, $O_3$, $NO_3$ radical, OH radical, $PM_{2.5}$, inorganic nitrate,sulfate, ammonium concentration changes between 2020 lockdown and 2020 BAU (no lockdown) scenario ($fX_{(GC,emi)}$). Error bars represent the 1 $\sigma$ of mean of ten metropolitan areas.

or both emission reductions for the 2020 lockdown period. The reduction in both emissions ($NO_X$ and VOC), suggested by Doumbia et al. (2021), results in slight increase in lockdown ozone levels ($< 2.5$ %) over only north-western Germany and slight decrease in lockdown ozone levels over other regions of Germany, compared to BAU levels. But, only reduction in $NO_X$ emission results in increased lockdown ozone levels (0-10 %) over all of Germany compared to BAU levels, which is also consistent with our results of increase in meteorology accounted for ozone levels over different metropolitan areas across

Germany during 2020 lockdown period compared to 2019 levels. This implies that VOC emissions were either not reduced at all or by a much smaller percentage than $NO_X$ emissions.

    The emission accounted for GC lockdown $NO_2$ concentrations decreased by 21 % ($fNO_{2(GC,emi)}$) while emission accounted for GC lockdown $O_3$ concentrations increased by 9 % compared to 2020 BAU (Fig. 3) . This is consistent with previous studies (such as Balamurugan et al. (2021); Gaubert et al. (2021)) which show that German metropolitan areas are in

a $NO_X$ saturated ozone production regime in spring. The emission accounted for GC lockdown PM concentrations show small decreases compared to 2020 BAU (Fig. 3). These results are consistent with previous studies (Gaubert et al., 2021; Hammer et al., 2021; Matthias et al., 2021; Menut et al., 2020), which used activity data to develop an emission reduction scenario and estimated small to no reduction in $PM_{2.5}$, a significant drop in $NO_2$ and marginal increase in $O_3$ levels during 2020 lockdown period, compared to BAU levels, over Northern-Europe including Germany.





We investigated the GC $PM_{2.5}$ composition for the studied period to determine the role of reduced $NO_X$ emission on total $PM_{2.5}$. Major secondary $PM_{2.5}$ components are nitrate, sulfate, ammonium and organic aerosol, which, on average, correspond to 24 %, 23 %, 15 % and 30 % of total $PM_{2.5}$, respectively, during March 21 to May 31, 2019 (Fig. C1). Mean relative contribution of $PM_{2.5}$ species for 2020 (BAU) and 2020 (lockdown) are shown in Fig. D1 and E1, respectively. The emission accounted for GC PM nitrate levels during the 2020 initial lockdown period (March 21 to April 30) are 9.5 % lower

than the 2020 BAU levels ($f\text{NIT}_{(GC,emi)}$) (Fig. 3 (a)), however, we see $NO_2$ decreased by 21 % during the same period. The decrease in emission accounted for GC PM nitrate is also less than the decrease in $NO_2$ during the second half of the lockdown (May 1 to May 31). The emission accounted for GC lockdown PM sulfate level show marginal increase (3.5 %), while emission accounted for GC lockdown PM ammonium shows marginal decrease (5.8 %), compared to 2020 BAU level. The slight increase (& decrease) in sulfate (& ammonium) was also found in the Hammer et al. (2021); Matthias et al. (2021)

studies, which used activity data to adjust the COVID-19 emission scenario.

    It is notable that the reduction in $NO_X$, a precursor to PM nitrate, does not directly translated into a decrease in PM nitrate formation. There are several pathways for the formation of nitric acid ($HNO_3$), which partition to PM nitrate (Allen et al., 2015; Bauer et al., 2007). The reaction of OH and $NO_2$ (homogeneous pathway) and the hydrolysis of $N_2O_5$ on aerosol particles (heterogeneous pathway) are the two major pathways (Chang et al., 2011, 2016; Mollner et al., 2010).

The reaction for $HNO_3$ formation via gas-phase oxidation of $NO_2$ by OH is:

$$NO_2 + OH \xrightarrow{\;M\;} HNO_3 \tag{R1}$$

The reactions resulting in $HNO_3$ formation via hydrolysis of $N_2O_5$ on aerosol surfaces are:

$$NO_2 + O_3 \longrightarrow NO_3 + O_2 \tag{R2}$$

$$NO_3 + NO_2 \xleftarrow{\;M\;} N_2O_5 \tag{R3}$$

$$N_2O_5 + H_2O(l) \longrightarrow 2\,HNO_3 \tag{R4}$$

    The formation of $HNO_3$ from the reaction of OH and $NO_2$ dominates during the day, while hydrolysis of $N_2O_5$ on aerosol particles dominates at night as OH night-time concentrations are low and $N_2O_5$ photolyzes easily (Russell et al., 1986). At

night, $NO_3$ radical can be an important precursor for PM nitrate via reactions (Eq. R3, R4) (Kang et al., 2021; Shah et al., 2020; Wang et al., 2013). The emission accounted for concentrations of OH and $NO_3$, which drive day and night-time formation of PM nitrate, increased substantially (15 % and 12 %, respectively) during the lockdown period compared to BAU (Fig. 3). The increase in OH radicals results from German metropolitan areas being in a $NO_X$ saturated regime (Shah et al., 2020). The increase in GC lockdown $NO_3$ levels is predominantly at night due to a significant increase in night-time $O_3$ (Fig. 4 (b,e)); the

reaction of $NO_2$ with $O_3$ is the most important source of $NO_3$ radical (Eq. R2) (Geyer et al., 2001).

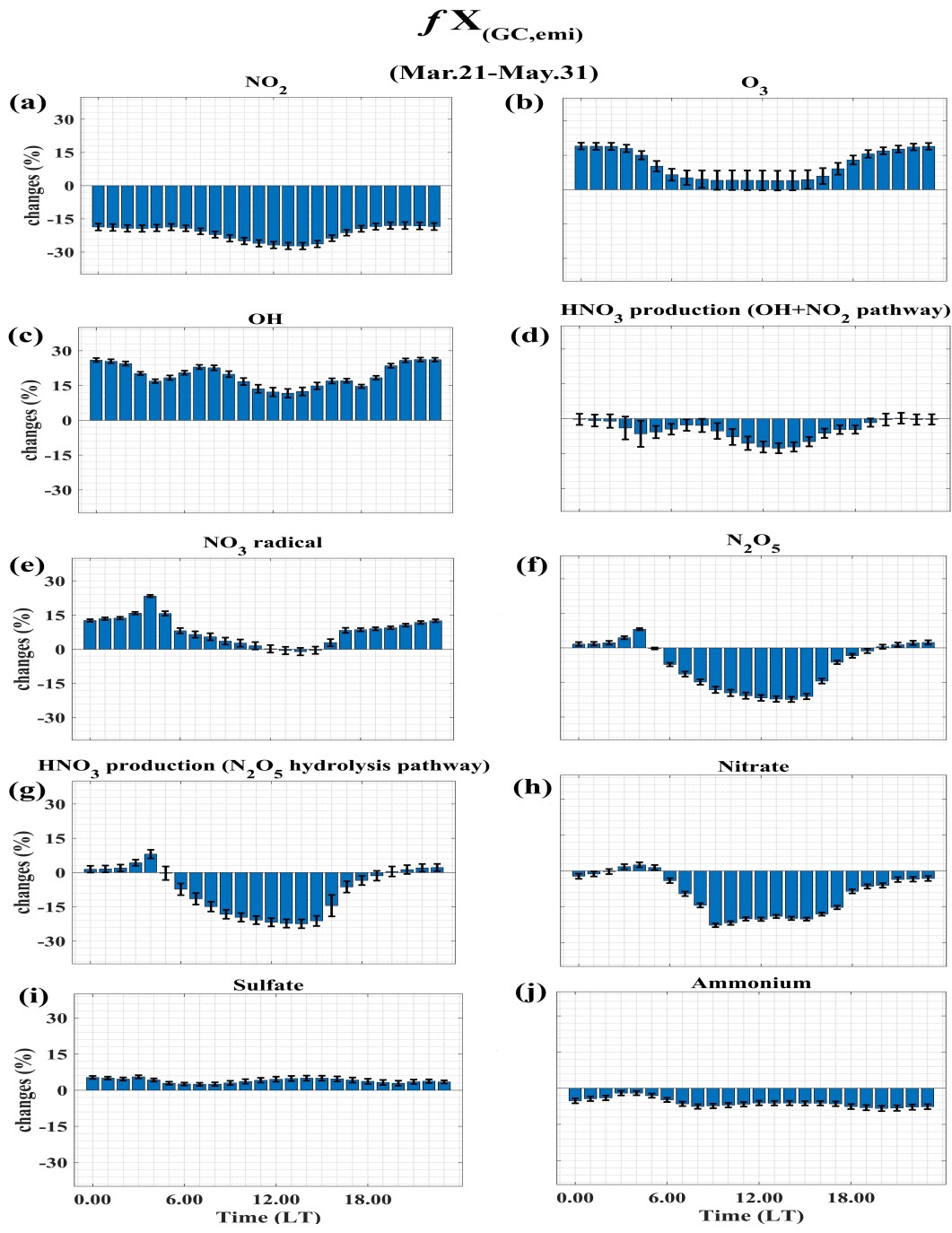

**Figure 4.** Diurnal cycle of emission accounted for GC $NO_2$, $O_3$, OH radical, $HNO_3$ production from oxidation of $NO_2$ by OH pathway, $NO_3$ radical, $N_2O_5$, $HNO_3$ production from $N_2O_5$ hydrolysis pathway, PM nitrate, sulfate, ammonium concentration changes between 2020 lockdown and 2020 BAU (no lockdown) scenario ($fX_{(GC,emi)}$). Error bars represent the standard error of respective hour in ten metropolitan areas.





Liu et al. (2020) have demonstrated that analyzing the diurnal cycle of total inorganic nitrate helps to identify the dominant pathway for the particulate nitrate production. The emission accounted for GC lockdown PM nitrate levels decreased significantly during the day, while night-time lockdown PM nitrate levels decreased slightly compared to BAU levels (Fig. 4 (h)). Even though GC lockdown OH levels increased, $HNO_3$ production from the $OH+NO_2$ reaction during the lockdown period is reduced due to significantly lower day-time $NO_2$ levels compared to BAU (Fig. 4 (d)); as a result, GC day-time lockdown PM nitrate levels are significantly lower compared to BAU levels. However, higher night-time $NO_3$ levels result in higher night-time $HNO_3$ production from $N_2O_5$ hydrolysis, resulting in slightly lower night-time lockdown PM nitrate compared to BAU (Fig. 4 (b,e,f,g)). This implies that the increase in $NO_3$ radical due to increased ozone partially offset the effect of reduced $NO_X$ on nitrate formation. Previous studies have also shown that $N_2O_5$ hydrolysis plays important role in nitrate formation than the gas-phase day-time pathway ($NO_2 + OH$) (Allen et al., 2015; Chan et al., 2021; Kim et al., 2014; Liu et al., 2020; Yan et al., 2019). Figure 5 illustrates the conceptual model of generalized day and night-time lockdown $NO_X$ chemistry compared to BAU scenario. The oxidation of $SO_2$ is a major source of sulfate, and the reaction with the OH radical dominates the gas-phase oxidation of $SO_2$ (Zhang et al., 2015). Therefore, the enhanced sulfate formation during the 2020 lockdown period could be due to the increased oxidizing capacity of atmosphere (OH) since we observe no significant change in emission accounted for GC $SO_2$ concentration, compared to BAU concentration (Fig. 3). Organic aerosol (OA) formation could be affected by the changes in oxidizing capacity of the atmosphere (Carlton et al., 2009), but no changes in emission accounted for GC lockdown OA were observed compared to 2020 BAU scenario. Therefore, the fact that no significant change in $PM_{2.5}$ due to lockdown restrictions is observed can be explained by a significant offset of the decreased day-time PM nitrate formation by enhanced formation of PM sulfate, while PM ammonium shows a marginal decrease.

## 4.3 Link between spring $PM_{2.5}$ pollution episodes and high $NH_3$ concentrations

It is worth noting that a significant fraction of $PM_{2.5}$ is PM nitrate. Ammonia ($NH_3$) is an important precursor for particulate nitrate formation (Ansari and Pandis, 1998; Banzhaf et al., 2013; Behera and Sharma, 2010; Wu et al., 2016). This explains the importance of monitoring and potentially regulating ammonia emissions. Therefore, the inter- and intra-annual changes in ammonia ($NH_3$) concentrations over Germany, as well as their relationship to $PM_{2.5}$ variability, are reviewed and analyzed further below. In Germany, atmospheric $NH_3$ levels follow a monthly pattern, with $NH_3$ levels peaking in April (Fig. 6 (b) and 7). $NH_3$ levels are also elevated during summer months. In Europe, major agricultural practices (fertilizer and manure applications) take place in the early spring (Petetin et al., 2016; Ramanantenasoa et al., 2018; Viatte et al., 2020). The higher atmospheric ammonia levels in April are attributable to agricultural practices such as fertilizer application. The high $NH_3$ values in summer are most likely due to warm climates (Kuttippurath et al., 2020). Monthly average $NH_3$ maps clearly show the high $NH_3$ values over North-West Germany from April to August, with particularly high values in April. It indicates that North-West Germany is a hotspot of ammonia emissions compared to the rest of the country. North-West Germany is known for its high livestock density (livestock farming (EUR, 2013; Scarlat et al., 2018)) and it is dominated by crop and grass land (ESA, 2017). Livestock farming and fertilizer application account for 75 % of $NH_3$ emissions in Europe (Webb et al., 2005). $NH_3$ concentrations in Germany vary greatly from year to year (inter-annual variabilities). We consider the period between



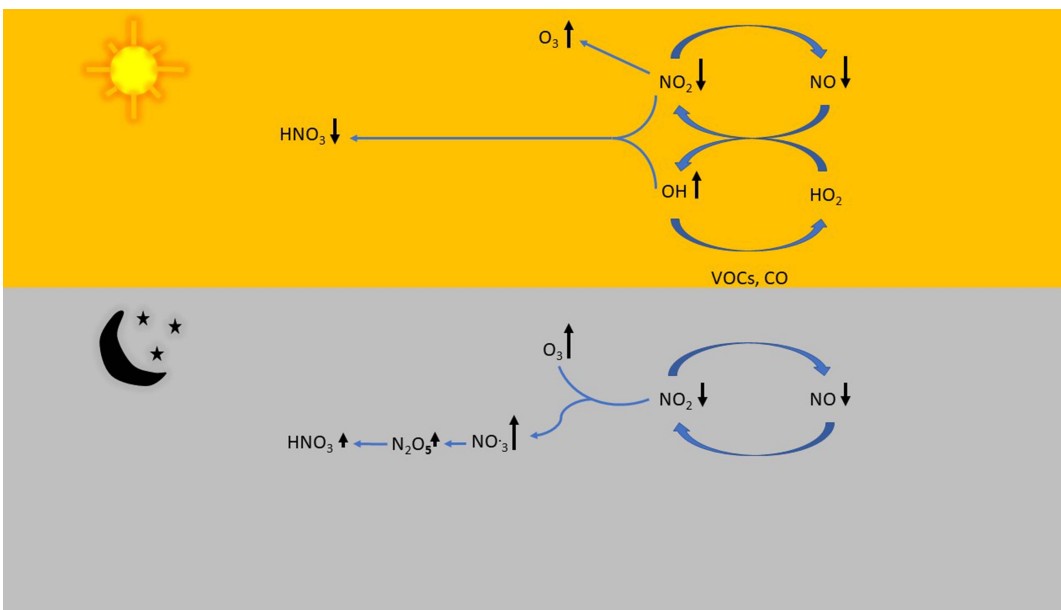

**Figure 5.** Generalized schematic diagram of day and night-time lockdown $NO_X$ chemistry compared to BAU scenario.

March 21 and April 30 when a stricter lockdown was in place to illustrate the inter-annual variability of atmospheric $NH_3$ between 2018 and 2020 (Fig. 8). $NH_3$ levels are lower in 2019 than in 2018, which can be attributed to the lower temperature in 2019 compared to 2018. Meanwhile, even though strict lockdown was in place, $NH_3$ levels in 2020 are higher than in 2019 and 2018, possibly due to low precipitation. High temperatures promote $NH_3$ volatilization (increases the $NH_3$ level in the atmosphere) (Ernst and Massey, 1960), whereas high rainfall favors wet deposition (removal of atmospheric $NH_3$). Schiferl

et al. (2016); Viatte et al. (2020) have also shown that meteorological parameters such as temperature and precipitation play a greater role in $NH_3$ inter-annual variability.

High PM pollution episodes are likely to occur frequently during the winter due to high residential heating demand and favorable meteorological conditions (e.g., low temperature and inversion condition). However, high concentrations of $PM_{2.5}$ are apparent in German metropolitan areas in the early spring (from the second half of March to the end of April, e.g., Fig. 6

(a) for Cologne metropolitan area). On March 21, 2020, the German government imposed COVID-19 lockdown restrictions. However, in-situ $PM_{2.5}$ concentrations during the initial lockdown period are higher than during the pre-lockdown period in 2020. High $PM_{2.5}$ levels from the second half of March to the end of April are also consistent with previous years without lockdown restrictions. It is notable that this high spring $PM_{2.5}$ episodes are associated with high $NH_3$ concentrations (Fig. 6 (b)). The high $PM_{2.5}$ events that occur in the spring have also been observed in other European cities, and they typically contain

ammonium nitrate and ammonium sulfate (Fortems-Cheiney et al., 2016; Renner and Wolke, 2010; Schaap et al., 2004; Viatte et al., 2020, 2021). Above, we show the high $NH_3$ levels in early spring (April) and summer months. High $PM_{2.5}$ concentrations are evident in spring, however, we did not observe high $PM_{2.5}$ episodes in summer (Fig. 6 (a)). It is also worth noting that even

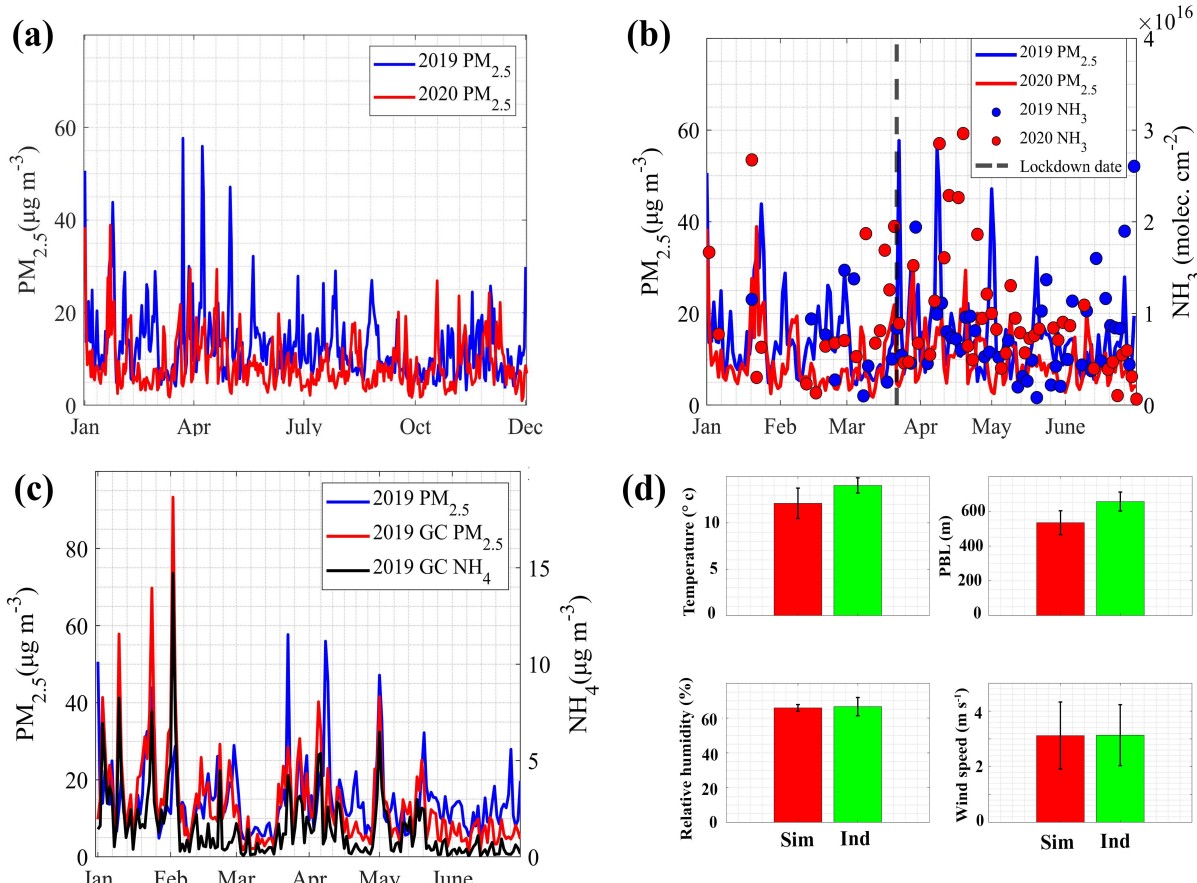

**Figure 6.** Annual daily mean $PM_{2.5}$ concentrations in Cologne (a). Zoom in closer to see high $PM_{2.5}$ pollution episodes with IASI $NH_3$ total columns in early spring in Cologne (b). Daily mean in-situ $PM_{2.5}$, GC simulated $PM_{2.5}$, and GC simulated $NH_4$ concentrations in Cologne (c). Statistical distribution of meteorological parameters for the cases "Simultaneous" (Sim) and "Independent" (Ind) in ten German metropolitan areas for 2018 and 2019 (d). "Simultaneous" - Simultaneous increase in $NH_3$ (IASI) and $PM_{2.5}$ (in-situ) concentrations on same day. "Independent" - Increase in $NH_3$ (IASI) concentration not corresponding to an increase in $PM_{2.5}$ (in-situ) concentration on same day. Error bars represent the $1\ \sigma$ of the mean of ten metropolitan areas (d).







Figure 7. Monthly mean IASI NH₃ total column at 1*1 degree resolution.



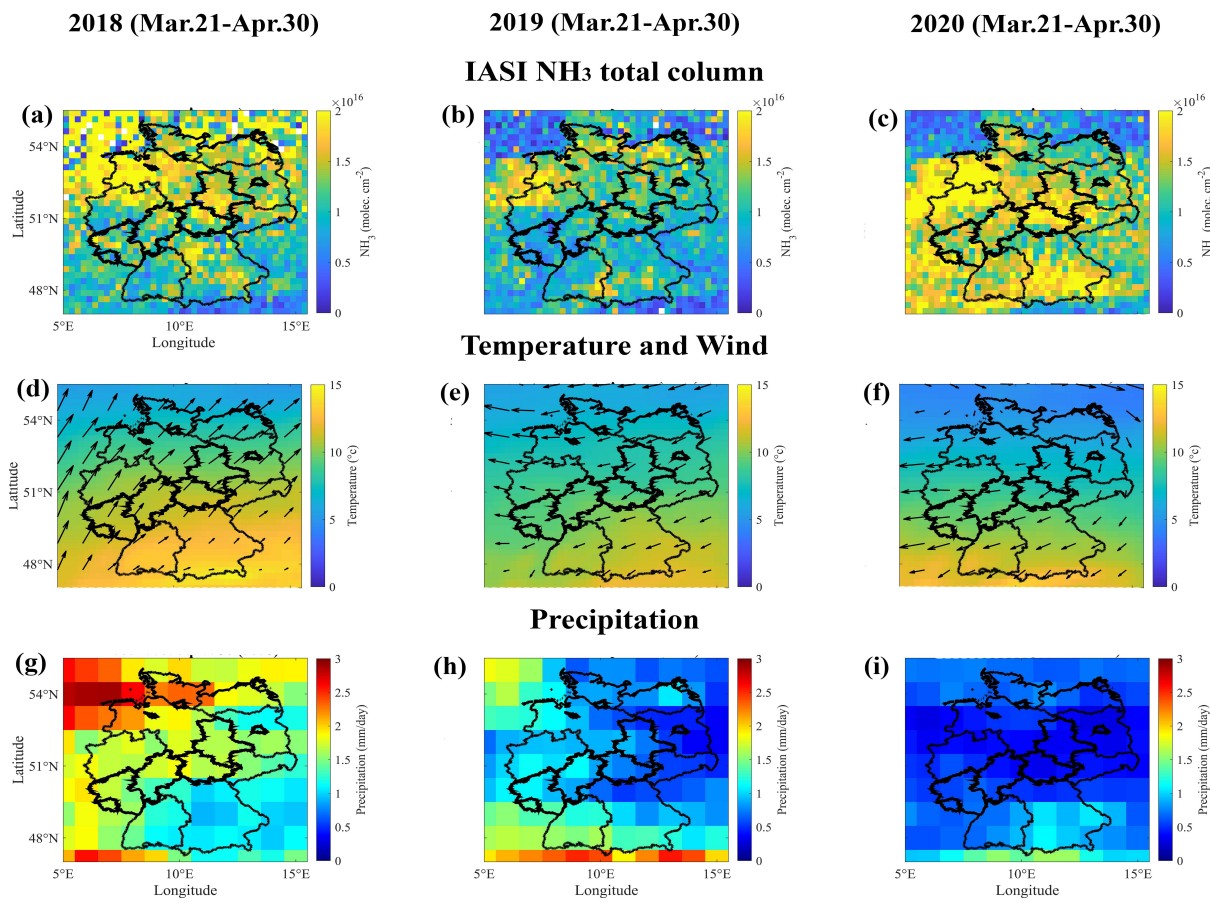

**Figure 8.** Mean IASI NH$_3$ total column (daily IASI NH$_3$ measurements gridded at 0.25 degree resolution) (top), mean temperature and wind (middle) and mean precipitation (bottom).

in the spring and winter PM$_{2.5}$ is not consistently high on days with high NH$_3$. This reflects the complexity of the process of gas to particle conversion. Despite high NH$_3$ concentrations, ammonia(NH$_3$)-to-ammonium(NH$_4$) conversion is mainly driven

by various meteorological factors such as temperature (and relative humidity). Studies (Viatte et al., 2020; Wang et al., 2015; Watson et al., 1994) have shown that conditions such as temperature of less than 10 °C and a high relative humidity of more than 70 % are optimal for atmospheric gas-phase NH$_3$ to transform into ammonium salts, mainly due to reversible ammonium nitrate formation, which depends on temperature and relative humidity; warm and dry conditions partition ammonia back to the gas phase (Mozurkewich, 1993). In comparison to summer, the impact of NH$_3$ on PM$_{2.5}$ formation is considerable for winter

and spring over Europe (Viatte et al., 2020, 2021) and the US (Schiferl et al., 2016). Summer weather is typically warmer (and has lower relative humidity) than winter and spring, which could explain why high NH$_3$ concentrations are not associated with high PM$_{2.5}$ in summer or late spring. To further demonstrate this for German metropolitan areas, we consider two cases ("Simultaneous" and "Independent") for 2018 and 2019 (Fig. 6 (d)). "Simultaneous" - Simultaneous increase in NH$_3$ (IASI)





and PM$_{2.5}$ (in-situ) concentrations on same day. "Independent" - Increase in NH$_3$ (IASI) concentration not corresponding to
an increase in PM$_{2.5}$ (in-situ) concentration on same day. As an example, for the Cologne metropolitan area, the temperature
and boundary layer height for the "Simultaneous" case (11.7±6.8 °C and 500.4±166.5 m, respectively) is lower than for the
"Independent" case (13.4±6 °C and 628.9±274.3 m, respectively). In addition to low temperature, low boundary layer height
results in higher pollutant concentrations and can thus result in more intense atmospheric chemical reactions. We found similar
results for other metropolitan areas, but with different absolute values (Fig. 6 (d)). The regional differences are unsurprising,
because other factors also influence the formation of PM$_{2.5}$ from NH$_3$ (e.g., other precursor concentrations such as NO$_X$ and
SO$_X$). However, these findings support previous studies and imply that low temperature and low boundary layer height are most
favorable for the formation of PM$_{2.5}$ during the periods of high NH$_3$. GC also simulates the high spring PM$_{2.5}$ concentrations
that have been observed, with high ammonium (NH$_4$) concentrations (Fig. 6 (c)).

## 5  Conclusions

Our study estimates the influence of anthropogenic emission reductions on PM$_{2.5}$ concentration changes during the 2020 lock-
down period in German metropolitan areas. Mean meteorology accounted for PM$_{2.5}$ concentrations decreased by 5 % during
the 2020 lockdown period (spring) compared to the corresponding period in 2019. However, during the 2020 pre-lockdown
period (winter), meteorology accounted for PM$_{2.5}$ concentrations are 19 % lower than in 2019. Meanwhile, meteorology ac-
counted for NO$_2$ levels decreased 23 % during the 2020 lockdown period, which is a larger decrease than 2020 pre-lockdown
period compared to 2019 (9 %). No significant change in meteorology accounted for SO$_2$ and CO concentrations were observed
during the 2020 lockdown period, compared to 2019.

The GC model with the COVID-19 emission reduction scenario based on observations (23 % reduction in NO$_X$ emission
with unchanged VOC and SO$_2$) supports our findings of only a marginal decrease in PM$_{2.5}$ and a significant decrease in NO$_2$
levels. Due to being in a NO$_X$ saturated ozone production regime, the GC lockdown OH and O$_3$ concentrations increased by 15
% and 9 %, respectively, compared to BAU levels. Despite an increase in OH radicals, the GC lockdown PM nitrate formation
decreased significantly during the day, due to a significant decrease in NO$_2$, compared to the BAU scenario. Increased night-
time ozone, however, results in increased night-time NO$_3$, despite decreased NO$_2$, in turn, resulting in slightly increased
night-time N$_2$O$_5$ concentration and only a small change in night-time PM nitrate. Overall this results in a small decrease
in daily PM nitrate. In addition, the increased OH concentration results in a marginal increase of sulfate formation. Nitrate,
sulfate, ammonium and organic aerosol are the major secondary components of PM$_{2.5}$. The decreased day-time PM nitrate is
partially offset by the enhanced PM sulfate, and there is no significant impact from slightly decreased PM ammonium and no
change in organic aerosol, resulting in a marginal decrease in PM$_{2.5}$ concentrations during the lockdown period.

Based on our findings, we suggest that additional emission control measures aimed at reducing ozone pollution be imple-
mented which should also help reduce PM. A concurrent reduction of NO$_X$ and VOCs emissions should occur. Otherwise,
ozone levels will rise as NO$_X$ emissions drop, increasing oxidizing capacity, until a NO$_X$ limited ozone production regime
is reached. We also addressed the annual spring PM$_{2.5}$ pollution episodes in German metropolitan areas, which are associ-





ated with high $NH_3$ concentrations. North-West Germany is a hot-spot of $NH_3$ emissions, primarily emitted from livestock farming and intensive agricultural activities (fertilizer application), with high $NH_3$ concentrations in the early spring and summer months. Winter and spring meteorological conditions are more favorable for $PM_{2.5}$ formation from $NH_3$ than summer.

Unsurprisingly, low temperature (and low boundary layer height) is shown to be a favorable meteorological condition for the formation of $PM_{2.5}$ from $NH_3$. Regulation of $NH_3$ emissions, primarily from agriculture, has the potential to reduce $PM_{2.5}$ pollution significantly in German metropolitan areas.

In this study, a COVID-19 emission reduction scenario was created using meteorology accounted for proxy pollutant concentration changes, assuming that observed proxy pollutant concentration changes are due to the combined direct effects of

emission and meteorology changes. Our GC modeling study work reflects the assumed direct relationship between changes in meteorology accounted for $NO_2$ concentration and changes in $NO_X$ emission. This work also shows a direct relationship between changes in meteorology accounted for $SO_2$ (and CO) concentration and changes in $SO_X$ (and CO) emission. However, due to the non-linear feedback system in atmospheric chemistry, this assumption should be investigated further. Because of their similar sources, we use CO concentration as a proxy for anthropogenic VOC concentration. However, this is debatable

because VOC is more reactive than CO. We call for further advancements in estimating the emission changes during the lockdown period, which would allow us to estimate the precise sensitivity of $PM_{2.5}$ to changes in emissions from various sources and comparison of VOC emission inventories with observations. This will help in the implementation of appropriate air quality regulation strategies in the future.

*Data availability.* Hourly measurements of in-situ $NO_2$, $O_3$, $PM_{2.5}$, $SO_2$ and CO data are downloaded from (https://discomap.eea.europa.eu/

map/fme/AirQualityExport.htm). The TROPOMI $SO_2$ data are obtained from https://s5phub.copernicus.eu/. The IASI $NH_3$ data are obtained from https://iasi.aeris-data.fr/catalog/. Hourly ERA5 meteorological data are available at https://cds.climate.copernicus.eu/.





**Figure A1.** Meteorology unaccounted for (red) and meteorology accounted for (green) mean changes in $PM_{2.5}$, $NO_2$, $SO_2$ and $O_3$ concentrations between 2020 and 2019 in ten German metropolitan areas. Error bars represent the 1 $\sigma$ of mean of ten metropolitan areas.



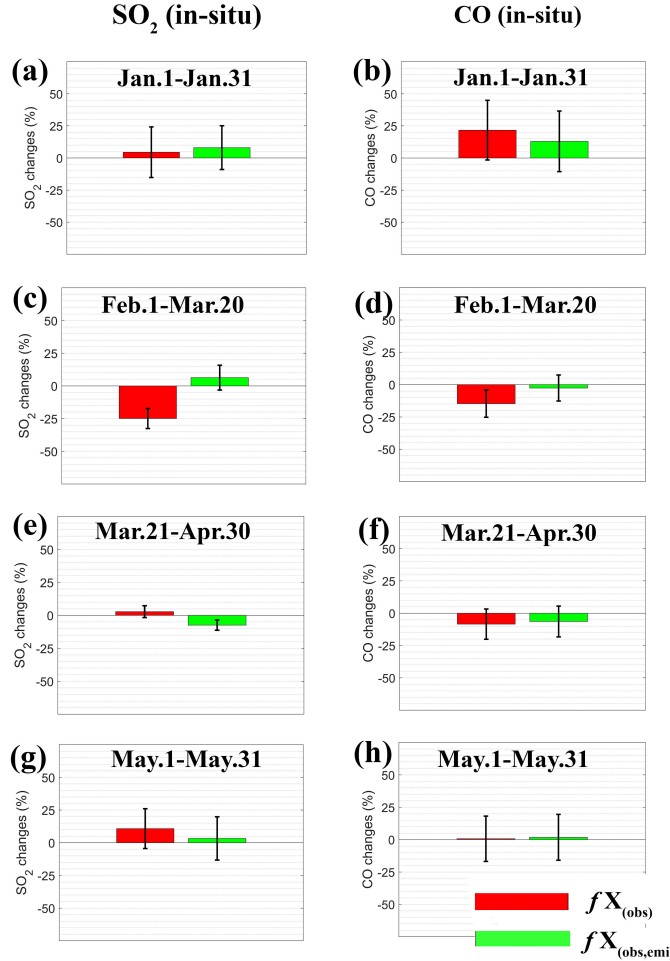

**Figure B1.** Meteorology unaccounted for (red) and meteorology accounted for (green) mean changes in in-situ $SO_2$ (Bremen, Dresden, Frankfurt, Hamburg and Leipzig) and in in-situ CO (Bremen, Frankfurt, Hamburg, Hanover, Munich and Stuttgart) between 2020 and 2019. Error bars represent the 1 $\sigma$ of mean of above mentioned metropolitan areas.





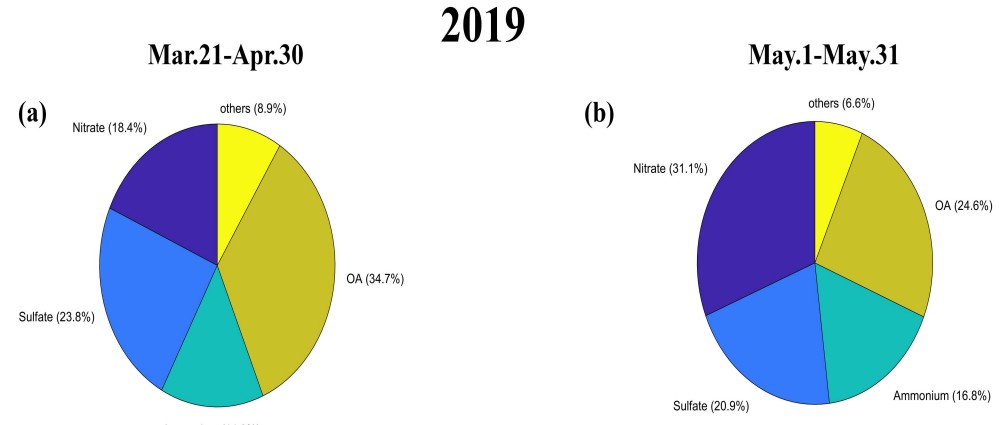

**Figure C1.** Mean relative contributions of PM$_{2.5}$ species simulated by GC for 2019.





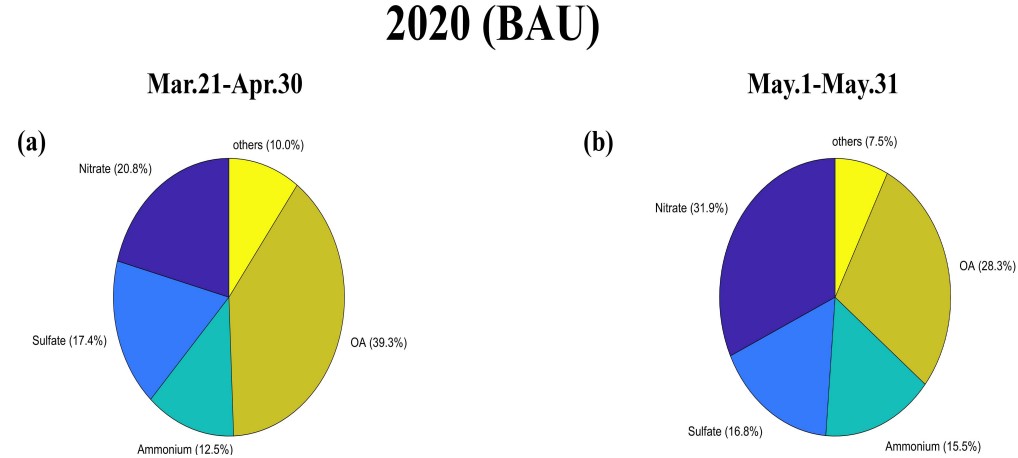

**Figure D1.** Mean relative contributions of $PM_{2.5}$ species simulated by GC for 2020 (no lockdown).



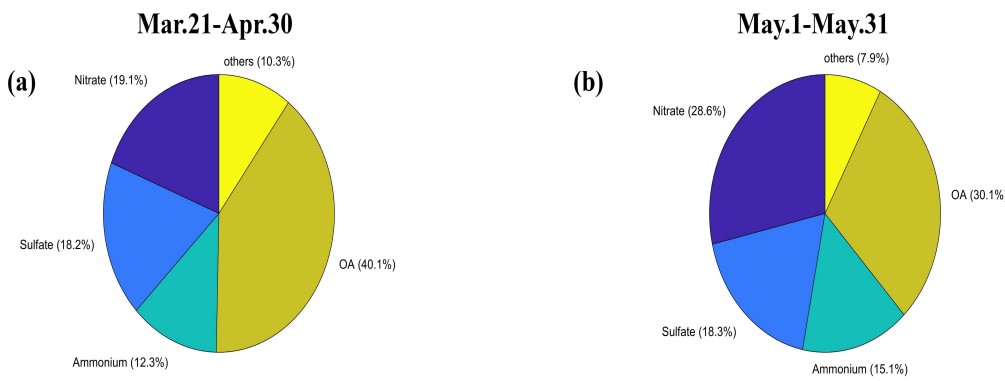

**Figure E1.** Mean relative contributions of PM$_{2.5}$ species simulated by GC for 2020 (lockdown).





*Author contributions.* ZQ performed the modeling work; VB, and XB obtained the measurement data; VB analyzed the data and wrote the manuscript draft; JC, and FK supervised the work and edited the manuscript.

*Competing interests.* The authors declare that they have no conflict of interest.

*Acknowledgements.* Vigneshkumar Balamurugan, Jia Chen, and Xiao Bi are supported by Institute for Advanced Study, Technical University of Munich, through the German Excellence Initiative and the European Union Seventh Framework Program (Grant: 291763) and in part by the German Research Foundation (DFG) (Grant: 419317138). Frank N. Keutsch is funded and supported by the Harvard' Solar Geoengineering Research Program.





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
