# Peer review of "Secondary PM2.5 decreases significantly less than NO2 emission reductions during COVID lockdown in Germany"

_Atmospheric Chemistry and Physics, 2022_

## Author Comment (AC1)

Dear Reviewer,

We appreciate your comments and suggestions, which have helped us improve our manuscript further. We have made the necessary changes to the manuscript, which can be found in the attached file (Track Changes). The following is a response to your comments and suggestions. Corresponding changes in the revised manuscript are also made available below, if applicable, at the appropriate places.

Sincerely,

On behalf of all co-authors, Vigneshkumar Balamurugan
* * *
**The manuscript entitled "Secondary PM decreases significantly less than NO2 emission reductions during COVID lockdown in Germany" by Vigneshkumar Balamurugan et al. explored the drivers of slight decrease of PM2.5 compared to NO2 emission during COVID-19 lockdown in Germany. The manuscript provides valuable information for understanding PM pollution under rigorous emission reduction measures and efficiently directing PM mitigation in the future. It is recommended that this manuscript be reconsidered for publication after major revisions.**

Thank you so much for reading and reviewing our manuscript! We carefully reviewed and considered your comments/suggestions, and made improvements in the revised manuscript.

**General comments:**

**Line 54:" The composition of PM thus varies greatly depending on time and location; for example, in urban areas nitrate and organic aerosol often dominate in winter time". More cases should be given to support this sentence.**

We cited additional studies to support this statement.

| Lines 55-58: | *The composition of PM thus varies greatly depending on time and location; for example, in urban areas nitrate and organic aerosol often dominate in winter time (Cesari et al., 2018; JudaRezler et al., 2020; Samek et al., 2020; Salameh et al., 2015; Womack et al., 2019; Zhai et al., 2021).* |
|---|---|

**Line 133:" The fractional change in meteorology accounted for pollutant concentration between 2020 and 2019, i.e., pollutant concentration changes between 2020 and 2019 due to emission changes only" This definition is misleading. According to your definition of ΔPM2.5(obs) and ΔPM2.5(GC), the**

**ΔPM2.5(obs,emi) should be the change of PM2.5 caused only by emission. If so, relative descriptions in the whole paper should be revised correspondingly.**

$\Delta PM_{2.5(obs,emi)}$ = Absolute concentration changes (µg m$^{-3}$) after accounting for meteorology (caused only by emission) between 2020 and 2019.

$f\,PM_{2.5(obs,emi)}$ = Fractional concentration changes (%) after accounting for meteorology (caused only by emission) between 2020 and 2019.

We hope this clarifies your comment. We also made minor changes to the sentence and equation to make it clearer to the reader.

| Lines 139-141: | *The fractional change in meteorology accounted for pollutant concentration between 2020 and 2019, i.e., fractional change (%) in pollutant concentration between 2020 and 2019 due to emission changes only, is calculated as,*

$fPM_{2.5(obs,emi)} = (\Delta PM_{2.5(obs,emi)} / PM_{2.5(obs,2019)}) * 100$ |
|---|---|

**Line 170: We also compared the 2019 GC and 2019 observed in-situ PM2.5 concentrations and found that the GC and observed in-situ PM2.5 concentrations were in good agreement (R > 0.5 for all metropolitan areas, except Leipzig which has a R value of 0.39) (e.g.,Fig. 6 (c), for Cologne metropolitan area)." The performance of the model is the base of further analysis. Hence, more details of the statistical evaluation of the model performance for each site should be given. In addition, the agreement R is above 0.5 for most areas and is 0.39 for Leipzig. Personally, I think the R is not good enough.**

In the revised manuscript (Table A1), we now included the performance of the GC model for each metropolitan area. The R value in all cases is between 0.3 and 0.7, indicating moderate correlation (https://link.springer.com/article/10.1057/jt.2009.5).

Table A1. The statistical evaluation (R, RMSE and mean bias) of the GC model performance (PM$_{2.5}$) for the 2019 study period (January 1 to May 31).

| Metropolitan area | Correlation coefficient (R) | RMSE (µg m$^{-3}$) | Mean bias (GC – insitu / insitu) (%) |
|---|---|---|---|
| Bremen | 0.6 | 8.7 | -18.9 |
| Cologne | 0.5 | 11 | 11.7 |

| | | | |
|---|---|---|---|
| Dresden | 0.56 | 9.2 | -18.8 |
| Dusseldorf | 0.53 | 10.5 | -15.7 |
| Frankfurt | 0.58 | 9.3 | -37.4 |
| Hamburg | 0.67 | 8 | -12.7 |
| Hanover | 0.59 | 7.9 | -13.1 |
| Leipzig | 0.39 | 8.4 | -28.6 |
| Munich | 0.5 | 8.5 | -18.6 |
| Stuttgart | 0.53 | 8.6 | -16.1 |

**Line 273:" The increase in OH radicals results from German metropolitan areas being in a NOX saturated regime". From BAU to lockdown period, the meteorological condition changed, which could lead to higher temperature and higher solar radiation, and this has the potential to increase OH concentration. Hence, the influence of meteorological between different period in 2020 should be considered.**

Section 4.2 shows a comparison of 2020 (lockdown) and 2020 (no lockdown). The meteorology is the same in both cases; the only difference is in the emissions. As a result, the change in OH must be due to chemistry changes caused by changes in $NO_X$ emissions. We hope this clarifies your comment.

**Line 281:"However, higher night-time NO3 levels result in higher nighttime HNO3 production from N2O5 hydrolysis, resulting in slightly lower night-time lockdown PM nitrate compared to BAU" According to Figure 4, the change of nighttime HNO3 production from N2O5 hydrolysis is small compared to that during daytime. In addition, both of the production and sink of HNO3 should be considered to explain its influence on PM concentration.**

Thanks for pointing this out. We modified the sentence as follow,

| | |
|---|---|
| Lines 310-312: | *However, higher night-time $NO_3$ levels result in relatively unchanged night-time $HNO_3$ production from $N_2O_5$ hydrolysis, resulting in slightly lower night-time lockdown PM nitrate compared to BAU (Fig. 4 (b,e,f,g)).* |

Our GEOS-Chem simulation diagnostics does not allow us to extract the $HNO_3$ sink information directly. Deposition (dry and wet deposition) is the primary sink of $HNO_3$ in the troposphere. Since we discuss the difference between 2020 (lockdown) and 2020 (no lockdown), which have the same meteorology with different emission, we don't expect the rate of dry deposition (calculated based on different meteorological variables (http://wiki.seas.harvard.edu/geos-chem/index.php/Dry_deposition) and wet deposition to differ significantly.

**Specific comments:**

**The use of "emission accounted", and "meteorology accounted" makes the discussion part puzzled. The authors are suggested to use more clear phases.**

We agree with the reviewer that these words make the discussion a bit confusing sometimes. We discussed this with our other colleagues and chose these words, because they are more clear in terms of methodology than other words. However, we are open to choose if a reviewer is willing to suggest new words.

**Figure 1: The part of "Ground-truth measurements" is misleading, it should contain the observations data from 2019 and 2020.**

Thanks for the suggestion. We modified the figure 1, as you suggested.

[revised manuscript text omitted]

---

## Author Comment (AC2)

Dear Reviewer,

We appreciate your comments and suggestions, which have helped us improve our manuscript further. We have made the necessary changes to the manuscript, which can be found in the attached file (Track Changes). The following is a response to your comments and suggestions. Corresponding changes in the revised manuscript are also made available below, if applicable, at the appropriate places.

Sincerely,

On behalf of all co-authors, Vigneshkumar Balamurugan
* * *
**Review of "Secondary PM decreases significantly less than NO2 emission reductions during COVID lockdown in Germany" by Balamurugan et al. Built on their previous work, the authors investigated the role of anthropogenic emissions on PM2.5 changes during the COVID-19 lockdown in Germany. After subtracting the meteorological effects, they found that NOx emission decreased by about 20% but there were small changes in PM2.5 concentrations. By applying modeling analysis, they attributed the small decrease of PM2.5 to increased formation of sulfate and nighttime nitrate, offsetting the decreased formation of ammonium and daytime nitrate. In addition, the authors also discussed the role of NH3 emission in driving high PM2.5 episodes. Overall, the study provides some interesting results and adds insights in the formation of secondary aerosols. The methodology is reasonable and the manuscript is well-written. I think it fits well within the scope of ACP journal. I would suggest its acceptance after the following comments are well addressed.**

Thank you so much for taking the time to read and review our manuscript! We carefully reviewed and considered your comments/suggestions, and as a result, we improved the manuscript.

**Comments:**

**The study focused on PM2.5 only, so I would suggest to replace "PM" by "PM2.5" in the title.**

Thanks for the suggestion. The title of this paper is modified as you suggested,

| Title | *Secondary PM$_{2.5}$ decreases significantly less than NO$_2$ emission reductions during COVID lockdown in Germany* |
|-------|--------------------------------------------------------------------------------------------------------------------|

**It seems fine to fix anthropogenic emissions at 2014 in the simulations, but it will be better if the authors could add some discussion about the emission changes from 2014 to 2019.**

Thanks for the suggestion. In GEOS-Chem simulations, we used the 2014 CEDS anthropogenic emission inventory, the most recent version of which is 2014, with corresponding year's (2020 and 2019) natural and fire emissions and meteorology. In our study, however, we use the difference between two years (e.g., 2020 - 2019) or two cases (e.g., $2020_{lockdown}$ - $2020_{no\ lockdown}$). Therefore, the effects of anthropogenic emission changes between 2014 to 2019 or 2020 will be canceled out. We added the following sentences in the revised manuscript.

| Lines 121-124: | *Even though the 2014 CEDS anthropogenic emission inventory is used in GC simulations, the effects of anthropogenic emission changes between 2014 and 2019 or 2020 will be canceled out because we use the difference between two years (e.g., 2020 - 2019) or two cases (e.g., $2020_{lockdown}$ - $2020_{no\ lockdown}$) in our study.* |
| --- | --- |

**I am still concerned about the assumption of unchanged VOC emissions in response to COVID-19 lockdown, although the authors tried to justify this treatment in their reduction scenarios. If NOx emissions from transportation sector were strongly affected during the lockdown, there is a reduction in VOC emissions as well. What are the sectors mainly accounting for VOC emissions in Germany? More discussions are needed on this issue.**

According to the European Environment Agency (EEA) (https://www.eea.europa.eu/data-and-maps/indicators/eea-32-non-methane-volatile-1/assessment-4), the road transport sector accounts for 14.6 % of total NMVOC emissions, while the road transport sector accounts for 40.5 % of total $NO_X$ emissions (https://www.eea.europa.eu/data-and-maps/indicators/eea-32-nitrogen-oxides-nox-emissions-1/assessment.2010-08-19.0140149032-3#:~:text=EEA%2D33%20emissions%20of%20nitrogen,households'%20(13%25)%20sectors). According to Guevara et al. (2021), the transportation sector contributes nearly 90 % of the reduction in total anthropogenic $NO_X$ and VOC emissions during lockdown. Based on our assumption that meteorology accounted for $NO_2$ changes equal to $NO_X$ emission changes, we find that $NO_X$ has decreased by 23 %. Because the lockdown restrictions primarily reduced traffic-related emissions, we can directly extrapolate this to a reduction in road transportation-related emissions; approximately 43 % (23-40.50 / 40.50). This finding also corresponds to a 40 % decrease in traffic vehicle count (Gensheimer et al., 2021). Therefore, the decrease in VOC emission from transport sector should be 6 % (14.6 * 0.43). This value also corresponds to the estimated 7 % decrease in anthropogenic VOC emissions in Germany (Guevara et al., 2021). However, due to a significant decline in the transport sector's VOCs emission

in recent years, this reduction in VOC emission from the transport sector, calculated based on the EEA's 2015 data, should be even less than 6 %. There is also no evidence that lockdown measures affect the major source of VOC emissions, which are use of volatile chemical products such as cleaning agents and personal care products, as well as biogenic emissions. Because we use the relative difference between $2020_{lockdown}$ and $2020_{no\ lockdown}$, we expect that a decrease in total VOC emissions of less than 6 % will have no significant impact on the results.

| Lines 251-263: | *According to the European Environment Agency (EEA) (https://www.eea.europa.eu/data-and-maps/indicators/eea-32-non-methane-volatile-1/assessment-4), the road transport sector accounts for 14.6 % of total NMVOC emissions, while the road transport sector accounts for 40.5 % of total $NO_X$ emissions (https://www.eea.europa.eu/data-and-maps/indicators/eea-32-nitrogen-oxides-nox-emissions 1/assessment.2010-08-19.0140149032-3::text=EEA%2D33%20emissions%20of%20nitrogen,households'%20(13 %25)%20sectors). According to Guevara et al. (2021), the transportation sector accounts for nearly 90 % of the reduction in total anthropogenic $NO_X$ and VOC emissions during lockdown. As we noted that $NO_X$ emission decreased by 23 %, and the lockdown restrictions primarily reduced traffic-related emissions, we can directly extrapolate this to a reduction in road transportation-related emissions; approximately 43 % (23-40.50 / 40.50). This finding also corresponds to a 40 % decrease in traffic vehicle count (Gensheimer et al., 2021). Therefore, the decrease in VOC emission from transport sector should be 6 % (14.6 \* 0.43). However, due to a significant decline in the transport sector's VOC emission in recent years, this reduction in VOC emission from the transport sector, calculated based on the EEA's 2015 data, should be even less than 6 %. There is also no evidence that lockdown measures affect the major source of VOC emissions, which are use of volatile chemical products such as cleaning agents and personal care products, as well as biogenic emissions.* |
|---|---|

**The explanation of ozone increases is not quite clear. It is possible that ozone formation efficiency was increased in response to NOx reduction under NOx-saturated regime. However, this reason might not work both for daytime ozone and nighttime ozone. In the cold season, ozone could be strongly titrated by NOx emissions which maybe directly increase ozone at night. I would like the authors add some analysis on the changes of Ox (NO2+O2) that can be used to isolate the effect from weakened titration.**

Thank you for the suggestion. In the revised manuscript, we included the total oxidant ($O_X = O_3 + NO_2$) analysis. The included figure and discussions in the revised manuscript are given below,

**Lockdown period (Mar.21–May.31)**

[Figure]

[Figure]

Figure C1. Meteorology accounted for mean changes in in-situ $O_X$ between 2020 and 2019 (left). Diurnal cycle of emission accounted for GC $O_X$ concentration changes between 2020 lockdown and 2020 BAU (no lockdown) scenario (right). Error bars represent the 1 σ of mean of ten metropolitan areas.

| Lines 213-217: | *It is also possible that the increase in ozone is due to less ozone destruction via lower NO titration, in addition to an increase in ozone formation efficiency through $NO_X$ saturated regime chemistry. The meteorology accounted for mean $O_X$ (= $NO_2$ + $O_3$) concentrations in the 2020 lockdown period is 2 % higher than in 2019 (Fig. C1(a)), implies that the reduced NO titration effect partly contributed to the increased ozone. $O_X$ analysis also implies that the decrease in $NO_2$ was offset by an increase in $O_3$, and ozone production is overwhelmingly $NO_X$ saturated in Germany.* |
|---|---|

| Lines 267-268: | *However, the diurnal cycle of GC $O_X$ changes between 2020 lockdown and BAU suggests that night-time ozone increases are solely due to a decrease in NO titration effects (Fig. C1(b)).* |
|---|---|

**I am wondering if there is ambient measurement for PM2.5 components. It deserves a comparison between simulated and observed PM2.5 species concentrations.**

In Germany, seven measurement stations measure PM$_{2.5}$ components (nitrate, sulfate, organic carbon, and ammonium) and provide daily averaged concentrations. However, there are fewer than 28 days of measurement days available for six measurement stations during the 2019 study period. Therefore, we compared the 2019 GC-simulated nitrate and ammonium concentrations to data from another one urban station (it has no sulfate and organic carbon measurements). The results are included in the revised manuscript.

[revised manuscript text omitted]

---

## Author Comment (AC3)

Dear Reviewer,

We appreciate your comments and suggestions, which have helped us improve our manuscript further. We have made the necessary changes to the manuscript, which can be found in the attached file (Track Changes). The following is a response to your comments and suggestions. Corresponding changes in the revised manuscript are also made available below, if applicable, at the appropriate places.

Sincerely,

On behalf of all co-authors, Vigneshkumar Balamurugan
* * *
**The authors present measurements from ten metropolitan areas in Germany to evaluate the impact of lockdown restrictions on air pollutant concentrations. They use the GEOS-Chem (GC) chemical transport model to simulate the pollutant concentrations for 2020 and 2019 and derive the percent changes during the lockdowns to find that although NO2 reductions were evident PM concentrations did not drastically change. Furthermore, they discuss the impacts of the NOx reductions on radical and ozone concentrations as well as PM2.5 formation and the role of NH3 emissions on PM pollution. This paper is interesting and fits well within the scope of ACP after the following comments are answered.**

Thank you so much for reading and reviewing our manuscript!

**Main comments:**

**My main concern is on the assumption that the VOC emissions did not change during the lockdowns based on a limited number of published studies that only account for a small fraction of the VOCs. Given that VOCs can originate from multiple sources that vary by season and meteorology I consider that there is limited confidence in this assumption. Furthermore, VOCs will be responsible for SOA in the model and can account for a significant part of the PM mass. I consider that a sensitivity analysis of the model to VOC changes would be a more honest approach and valuable addition to this study. The response of SOA to these changes and their relative influence compared to NH3 emissions, especially during PM pollution days, would indicate whether VOCs are also an essential source of PM pollution in future scenarios.**

We agree with the reviewer that VOCs can originate from multiple sources, e.g., biogenic VOC emission, which is a major source of VOC emission, varies by season and meteorology. Section 4.2 presents a comparison of 2020 (lockdown) and 2020 (no lockdown). The meteorology is the same in both cases. We also use the 2020 natural and

fire emission in GC simulations. Because we calculate the relative difference between $2020_{lockdown}$ and $2020_{no\ lockdown}$, any change in biogenic and natural VOC emission has no effect.

According to the European Environment Agency (EEA) (https://www.eea.europa.eu/data-and-maps/indicators/eea-32-non-methane-volatile-1/assessment-4), the road transport sector accounts for 14.6 % of total NMVOC emissions, while the road transport sector accounts for 40.5 % of total $NO_X$ emissions (https://www.eea.europa.eu/data-and-maps/indicators/eea-32-nitrogen-oxides-nox-emissions-1/assessment.2010-08-19.0140149032-3#:~:text=EEA%2D33%20emissions%20of%20nitrogen,households'%20(13%25)%20sectors). According to Guevara et al. (2021), the transportation sector contributes nearly 90 % of the reduction in total anthropogenic $NO_X$ and VOC emissions during lockdown. Based on our assumption that meteorology accounted for $NO_2$ changes equal to $NO_X$ emission changes, we find that $NO_X$ has decreased by 23 %. Because the lockdown restrictions primarily reduced traffic-related emissions, we can directly extrapolate this to a reduction in road transportation-related emissions; approximately 43 % (23-40.50 / 40.50). This finding also corresponds to a 40 % decrease in traffic vehicle count (Gensheimer et al., 2021). Therefore, the decrease in VOC emission from transport sector should be 6 % (14.6 * 0.43). This value also corresponds to the estimated 7 % decrease in anthropogenic VOC emissions in Germany (Guevara et al., 2021). However, due to a significant decline in the transport sector's VOCs emission in recent years, this reduction in VOC emission from the transport sector, calculated based on the EEA's 2015 data, should be even less than 6 %. There is also no evidence that lockdown measures affect the major source of VOC emissions, which are use of volatile chemical products such as cleaning agents and personal care products, as well as biogenic emissions. Because we use the relative difference between $2020_{lockdown}$ and $2020_{no\ lockdown}$, we expect that a decrease in total VOC emissions of less than 6 % will have no significant impact on the results.

| Lines 251-263: | *According to the European Environment Agency (EEA) (https://www.eea.europa.eu/data-and-maps/indicators/eea-32-non-methane-volatile-1/assessment-4), the road transport sector accounts for 14.6 % of total NMVOC emissions, while the road transport sector accounts for 40.5 % of total $NO_X$ emissions (https://www.eea.europa.eu/data-and-maps/indicators/eea-32-nitrogen-oxides-nox-emissions-1/assessment.2010-08-19.0140149032-3::text=EEA%2D33%20emissions%20of%20nitrogen,households'%20(13%25)%20sectors). According to Guevara et al. (2021), the transportation* |
|---|---|

> *sector accounts for nearly 90 % of the reduction in total anthropogenic $NO_X$ and VOC emissions during lockdown. As we noted that $NO_X$ emission decreased by 23 %, and the lockdown restrictions primarily reduced traffic-related emissions, we can directly extrapolate this to a reduction in road transportation-related emissions; approximately 43 % (23-40.50 / 40.50). This finding also corresponds to a 40 % decrease in traffic vehicle count (Gensheimer et al., 2021). Therefore, the decrease in VOC emission from transport sector should be 6 % (14.6 \* 0.43). However, due to a significant decline in the transport sector's VOC emission in recent years, this reduction in VOC emission from the transport sector, calculated based on the EEA's 2015 data, should be even less than 6 %. There is also no evidence that lockdown measures affect the major source of VOC emissions, which are use of volatile chemical products such as cleaning agents and personal care products, as well as biogenic emissions.*

We agree with the reviewer that a sensitivity analysis of changes in VOC emission on PM formation is an important study with significant implications. This sensitive study, we believe, should be conducted separately, taking into account the impact of different ranges of changes in $NO_X$ and VOC emissions on PM formation. But, in our study, we believe that reducing total VOC emissions by 6 % will have no discernible effect on the outcomes.

**Given that this work is based on the WRF model it would be great to see a more detailed evaluation of the model for the different gas- and particle-phase components. Evaluation of the model at high and low concentration periods from previous years and how accurately they are predicted would be of value and give some context on the uncertainty of this approach. Evaluation of the chemical composition derived by the model to ambient observations would also be important. Are there any chemically speciated measurements in Germany during this period that the authors could compare their model to? If not, has this been done in the past and what was the agreement of the model to the observations?**

The performance of the GEOS-Chem model ($PM_{2.5}$) for each metropolitan area for the 2019 study period is now included in the revised manuscript (Table A1). In our previous work (Balamurugan et al. 2021), we compared $NO_2$ and $O_3$ measurements from GEOS-Chem and in-situ for the same period (2019), and showed that they were in good agreement.

In Germany, seven measurement stations measure $PM_{2.5}$ components (nitrate, sulfate, organic carbon, and ammonium) and provide daily averaged concentrations. However,

there are fewer than 28 days of measurement days available for six measurement stations during the 2019 study period. Therefore, we compared the 2019 GC-simulated nitrate and ammonium concentrations to data from another one urban station (it has no sulfate and organic carbon measurements). The results are included in the revised manuscript.

| | |
|---|---|
| Lines 181-183: | *We also compared the 2019 GC simulated nitrate and ammonium concentration for the urban measurement station in Germany (14.33°E, 51.75°N). The statistical evaluation (R, RMSE and mean bias) of the model performance is given in Table B1.* |

Table B1. The statistical evaluation (R, RMSE and mean bias) of the GC model performance (nitrate and ammonium in PM2.5) for the 2019 study period (January 1 to May 31). For this comparison, data from the urban measurement station (14.33°E, 51.75°N) is used.

| Species | Correlation coefficient (R) | RMSE ($\mu g\ m^{-3}$) | Mean bias (GC – insitu / insitu) (%) |
|---|---|---|---|
| Nitrate | 0.51 | 2.33 | -32.1 |
| Ammonium | 0.45 | 1.34 | 37 |

**Other comments:**

**Line 51: First time that VOCs are introduced**

Thanks for pointing this out. We now included the full form of abbreviation in the text.

| | |
|---|---|
| Lines 50-54: | *PM sources include both direct/primary sources (vehicle and industrial emissions, wind-blown dust, pollen, wildfires, etc.) as well as secondary formation (gas-to-particle conversion process) via atmospheric chemical reaction of precursor compounds such as $NO_X$ (nitrogen oxides), $SO_2$ (sulfur dioxide), $NH_3$ (ammonia), VOCs (Volatile Organic Compounds) and other* |

| | *organic compounds, including compounds that have partitioned from primary aerosol back to the gas-phase, followed by partitioning to the condensed phase.* |
|---|---|

**Line 225: Which VOCs? How much of the reactivity do they represent?**

It is intended to be "anthropogenic VOCs". Those studies did not provide VOC reactivity information; rather, they provided total changes in anthropogenic VOC emission from various anthropogenic emission sectors.

| Lines 238-240: | *For those studies there are large differences in estimated anthropogenic VOC emission changes for Europe; Doumbia et al. (2021) estimated 34 % while Guevara et al. (2021) estimated 8 % reduction in anthropogenic VOC emissions.* |
|---|---|

**Line 235: What are the expected VOC emissions during the winter in Europe?**

Line 235 (unrevised manuscript): *This implies that VOC emissions were either not reduced at all or by a much smaller percentage than $NO_X$ emissions.*

This is intended to be "This implies that anthropogenic VOC emissions were either not reduced at all or by a much smaller percentage than $NO_X$ emissions, compared to the BAU scenario".

In Europe, anthropogenic VOC emissions (Ethane and Propane) dominate in the winter, while biogenic VOCs (Isoprene, Pinene) have a low contribution because they are primarily driven by temperature and solar radiation (Debevec et al., 2021). However, our study period focuses on spring, when biogenic VOCs and oxygenated VOCs are also important.

Based on previous studies and observed increase in ozone, we justify that anthropogenic VOC emissions were either not reduced at all or by a much smaller percentage than $NO_X$ emissions during the 2020 lockdown compared to 2020 BAU. Because we use the relative difference between $2020_{lockdown}$ and $2020_{no\ lockdown}$, any change in biogenic and natural VOC emission has no effect because both cases consider the same meteorology and time period. We also modified this sentence to make it more clear, as follows,

| Lines 249-250: | *This implies that anthropogenic VOC emissions were either not reduced at all or by a much smaller percentage than anthropogenic NO$_X$ emissions, compared to the BAU scenario.* |
|---|---|

**Line 290-294: OA formation and specifically SOA could also be affected by changes in VOC emissions both of biogenic and anthropogenic nature. Further discussion here would be of value.**

We agree with the reviewer that changes in primary/secondary biogenic and anthropogenic sources could have an impact on organic aerosol, specifically SOA. However, because we assume no changes in VOC emissions and compare the 2020 $_{lockdown}$ and 2020$_{no\ lockdown}$, we limit our discussion in section 4.2 that changes in the atmosphere's oxidizing capacity may affect OA formation. To make it clear, in the conclusion section, we added the following sentence.

| Lines 36-50: | *Organic aerosol accounts for nearly 30 % of total PM$_{2.5}$, which could be influenced by both primary/secondary biogenic and anthropogenic sources. However, our study is limited to examining the effects of NO$_X$ emission changes on PM$_{2.5}$ formation. Therefore, future studies on VOC emission changes on OA formation during high PM pollution episodes, particularly in the spring, will be more important in mitigating PM pollution.* |
|---|---|

**Line 335-337: I find this statement a stretch given the number of other sources of PM pollution.**

We added the following sentences in the revised manuscript.

| Lines 365-367: | *Furthermore, It is important to note that PM$_{2.5}$ anthropogenic precursor emissions (NO$_X$, SO$_2$, VOCs) have a seasonal cycle, with higher emissions in winter than summer; however, biogenic VOC emissions dominate in the summer.* |
|---|---|

**Line 339-348: Some statistics on how many days were the "simultaneous" or "independent" would be great here not only for one region but for all regions in Germany.**

Thanks for the suggestion. We included meteorological parameter statistics (and considered days) for the "simultaneous" or "independent" cases for all ten metropolitan areas in the revised manuscript (Table C1).

Table C1. The Statistical distribution of meteorological parameters for the cases "Independent" (each row top) and "Simultaneous" (each row bottom) in ten German metropolitan areas for 2018 and 2019.

| Metropolitan area | Days | Wind speed (m/s) | Temperature (° C) | Relative humidity (%) | Boundary layer height (m) |
|---|---|---|---|---|---|
| Bremen | 17 | 4.3 ± 2.1 | 13.6 ± 5.8 | 62.3 ±14.1 | 625.5 ± 211.1 |
| | 27 | 4.5 ± 2 | 11.5 ± 7 | 67.3 ± 16 | 541 ± 212.5 |
| Cologne | 16 | 3±2.2 | 13.4±6.1 | 74.3±11.4 | 628.9±274.31 |
| | 24 | 3.2±1.7 | 11.7±6.8 | 65.3±14.4 | 500.4±166.4 |
| Dresden | 24 | 1.9±1.1 | 14.9±6.9 | 68.6±12.8 | 578.9±220.7 |
| | 20 | 2.4±0.8 | 11.1±7.4 | 66.3±11 | 592.1±208.8 |
| Dusseldorf | 10 | 3.4±2.1 | 13.2±4.8 | 69±11.3 | 732.1±311.8 |
| | 30 | 3.4±1.8 | 13.5±5.6 | 66.2±13.5 | 494±168 |
| Frankfurt | 18 | 3.2±1.8 | 13.1±6.3 | 64.9±13.2 | 695.2±284.1 |
| | 21 | 2.2±1.1 | 13.1±6.6 | 63.6±13.6 | 442.8±194.5 |
| Hamburg | 14 | 5.4±2.5 | 13.7±6.5 | 57.5±11.8 | 705.3±249.2 |
| | 27 | 5.2±2.3 | 11.1±3.3 | 67.7±15 | 674.1±262 |
| Hannover | 14 | 3.2±2 | 14.2±7.8 | 62.5±10.4 | 697.5±210.2 |
| | 24 | 3.8±1.9 | 9.3±7.6 | 67.6±13.1 | 557.5±176.3 |

| | | | | | |
|---|---|---|---|---|---|
| Leipzig | 18 | 2.9±1.4 | 14.9±8 | 63.7±12.7 | 674.6±206.3 |
| | 30 | 3.4±1.6 | 11.2±7.1 | 61.9±10.8 | 532.3±227.3 |
| Munich | 26 | 2±1.1 | 15.5±5.4 | 71.5±12.3 | 599.8±196.3 |
| | 17 | 1.6±0.8 | 14.8±8.3 | 65.4±9.8 | 557.9±193.4 |
| Stuttgart | 22 | 1.9±0.9 | 13.8±6.4 | 71.7±11 | 600.7±234.9 |
| | 22 | 1.5±0.6 | 13.7±6.3 | 67.3±12.9 | 449±191.1 |

**Figure 6: I find this figure hard to follow and the messages are not clear to me. It would be great if the timeseries panels fit the whole page and the "simultaneous" or "independent" periods are highlighted by the background color of the graphs. Adding the temperature and RH timeseries would be great too. The authors can also include the NH3 measurements in a different panel and the background colors could guide the reader's eye's to evaluate whether there is a good or bad agreement between PM, NH3, RH, and temperature increases. Furthermore, it would be great to see a graph that highlights what happens in different regions of Germany and some more statistics on these trends to evaluate their importance.**

Thanks for the suggestion. We modified the figure as you suggested. We also included the statistics for all ten metropolitan areas in Table C1.

[Figure]

Figure 8. 2019 and 2020 annual daily mean in-situ PM₂.₅ concentrations in Munich (a). In figure panel (a), the vertical dashed line denotes the start of 2020 lockdown. 2019 daily mean in-situ PM₂.₅ and column NH₃ from IASI satellite (b, top). 2019 daily mean temperature and relative humidity (b, middle). 2019 daily mean wind speed and boundary layer height (b, bottom). The corresponding days for the cases "Simultaneous" are shaded with gray color, and for the cases "Independent" are shaded with cyan color. "Simultaneous" - Simultaneous increase in NH₃ (IASI) and PM₂.₅ (in-situ) concentrations on the same day. "Independent" - Increase in NH₃ (IASI) concentration not corresponding to an increase in PM2.5 (in-situ) concentration on the same day.

* * *

[revised manuscript text omitted]